# Learning Fingerprints for Medical Time Series with Redundancy-Constrained Information Maximization

Huayu Li [* 1]   ZhengXiao He [* 1]   Xiwen Chen [2]   Jingjing Wang [3]   Siyuan Tian [4]   Jinghao Wen [5]   Ao Li [1]

## Abstract

Learning useful representations from medical time series (MedTS), such as ECG and EEG, remains difficult due to high dimensionality, variable length, and noise. Existing self-supervised methods, including Masked Autoencoders, often produce pooled or sequence-level embeddings whose latent structure is not explicitly controlled. We propose TS-Fingerprint, a framework that compresses a variable-length MedTS input into a fixed-size set of $k$ latent Fingerprint Tokens. TS-Fingerprint uses a cross-attention bottleneck to generate these tokens and is trained with a dual-objective loss combining reconstruction and diversity regularization. The reconstruction objective preserves information needed to recover the input, while a Total Coding Rate (TCR)-based regularizer reduces redundancy among tokens. From a rate-distortion perspective, this design balances information preservation and compact, diverse coding. Experiments on ECG, EEG, and activity-recognition benchmarks show that TS-Fingerprint improves downstream classification while yielding decorrelated and functionally specialized tokens that support token-level interpretation.

## 1. Introduction

The efficacy of downstream clinical diagnosis hinges on representation learning, the ability to distill high-dimensional, noisy Medical Time Series (MedTS) into compact and meaningful latent codes. Recently, this domain has been dominated by the masked reconstruction paradigm adapted from computer vision (He et al., 2022). While these models (e.g.,

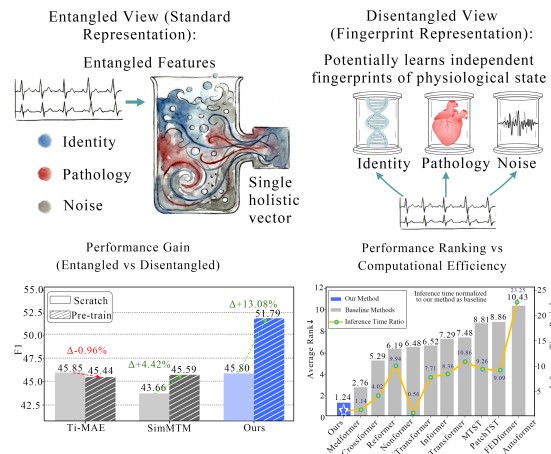

*Figure 1.* **Overview of TS-Fingerprint.** Standard MAE-style representations often require pooling over variable-length token sequences, whereas TS-Fingerprint compresses MedTS into a fixed set of fingerprint tokens with reduced redundancy. Bottom left compares pre-training gains against Ti-MAE and SimMTM. Bottom right reports average rank across datasets and metrics, together with normalized inference time.

SimMTM, TiMAE) excel at pre-training weights by forcing networks to memorize signal details (Dong et al., 2023), we question the clinical validity of this dominant paradigm. By prioritizing raw data fidelity (minimizing Mean Squared Error), these methods inadvertently produce entangled representations where critical physiological factors such as cardiac rhythm versus morphology are inextricably mixed across high-dimensional vectors. We argue that the failure lies not in the capacity of deep learning, but in the improper formulation of the learning objective: treating the latent space as a passive compression bucket rather than an active factorization engine.

In high-stakes medical environments, a representation that is merely sufficient for reconstruction but opaque in structure is untenable. An ideal clinical representation must be: (1) **Compressed** into a fixed-size vector invariant to input length (2) **Sufficient** in retaining diagnostic signals and (3) **Disentangled**, such that statistically independent latent dimensions correspond to distinct physiological factors. Existing general-purpose time-series learners largely fail

[1]University of Arizona, AZ, USA [2]Morgan Stanley, NY, USA [3]Clemson University, SC, USA [4]Microsoft Research, Beijing, China [5]Villanova University, IL, USA. Correspondence to: Huayu Li, Xiwen Chen <hl459@arizona.edu, xiwen.chen@morganstanley.com>.

*Proceedings of the 43rd International Conference on Machine Learning*, Seoul, South Korea. PMLR 306, 2026. Copyright 2026 by the author(s).

to meet these criteria, yielding holistic embeddings where digital biomarkers are diffused across the entire latent space.

To address these pitfalls, we fundamentally repurpose the goal of the bottleneck. As illustrated in Figure 1, we propose a paradigm shift from the **Entangled View**, which crams all signal variance (noise, identity, pathology) into a holistic vector to the **Fingerprint View**. By treating the encoder as a **Redundancy-Constrained Information Maximizer**, we assign $k$ fixed slots (Fingerprints) and force them to compete for signal variance. This imposes a structural bottleneck that encourages distinct tokens to specialize in different dominant modes of variation, thereby reducing representational redundancy. We demonstrate empirically that this inductive bias has the potential to isolate distinct physiological states in latent space.

Technically, we frame this as a **Disentangled Rate-Distortion** problem. We propose **TS-Fingerprint**, a framework designed as the practical realization of this objective. Our architecture compresses variable-length dynamics into a fixed set of $k$ semantically meaningful *Fingerprint Tokens*. This is achieved via a dual-objective optimization: a masked reconstruction loss that enforces sufficiency (maximizing mutual information $I(X; F')$) and a Total Coding Rate (TCR) regularizer that acts as a repulsive force, geometrically pushing the tokens to be statistically orthogonal.

Our main contributions are summarized as follows: **(1) Theoretical Formulation:** We formulate the MedTS representation problem within a Disentangled Rate-Distortion framework, demonstrating theoretically that minimizing latent redundancy leads to strictly superior sample efficiency for downstream clinical tasks **(2) Fingerprint Token Architecture:** We propose a novel encoder that learns a compact, fixed-size set of disentangled Fingerprint Tokens, providing a transparent and clinically interpretable alternative to holistic black-box embeddings **(3) SOTA Performance & Interpretability:** Extensive experiments across ECG, EEG, and activity benchmarks demonstrate that our method achieves state-of-the-art classification performance (Figure 1) while enabling precise, medically relevant explanations through token-specific attention maps.

## 2. Related Works

**Time-Series Backbones and Medical Representation Learning** Early medical time-series models relied on recurrent architectures such as LSTMs and GRUs (Cho et al., 2014), while recent work has increasingly adopted Transformers (Vaswani et al., 2017) for modeling long-range dependencies. To improve efficiency and generalization, many time-series backbones have been proposed, including patch-based models such as PatchTST (Nie, 2022), channel-wise architectures such as iTransformer (Liu et al., 2023),

convolutional models such as ModernTCN (Luo & Wang, 2024), and multi-resolution variants such as MTST (Zhang et al., 2024) and CSFormer (Wan et al., 2025). Medical time series introduce additional challenges, including noise, high dimensionality, and multi-scale physiological dynamics (Ismail Fawaz et al., 2020; Wang et al., 2024b). Specialized models such as Medformer (Wang et al., 2024b) address these issues through multi-granularity patching, while MTST (Zhang et al., 2024) uses multi-resolution attention to capture diverse temporal and frequency patterns. Although these models are effective for forecasting and classification, they mainly optimize prediction or reconstruction performance and typically output sequence-level features whose size depends on the input resolution or patching scheme. In contrast, our work focuses on learning a compact, fixed-size representation for medical time series. Rather than designing a more complex temporal backbone, we study whether a token bottleneck with redundancy regularization can produce representations that remain predictive while being easier to analyze at the token level.

**Disentangled and Redundancy-Reduced Representations** Disentangled representation learning seeks to separate factors of variation in the latent space, but recovering true independent generative factors is difficult without strong assumptions or supervision (Wang et al., 2024a). Recent self-supervised methods therefore often target weaker, measurable objectives such as redundancy reduction, decorrelation, or variance preservation. Examples include Barlow Twins (Zbontar et al., 2021), VICReg (Bardes et al., 2021), and maximal coding-rate reduction ($\text{MCR}^2$) (Yu et al., 2020). TS-Fingerprint is related to this line of work, but applies a coding-rate-inspired diversity objective across a small set of latent tokens, rather than across dimensions of a single global embedding. This encourages the tokens to carry complementary information while preserving a fixed-size interface for classification and token-level analysis.

**Self-Supervised Representation Learning** Self-supervised learning has been widely used to learn representations from unlabeled data. Contrastive methods, including SimCLR (Chen et al., 2020), TS2Vec (Yue et al., 2022), and TS-TCC (Eldele et al., 2021), learn invariance by contrasting augmented views, but often depend on augmentation design and negative sampling. Generative methods such as Masked Autoencoders (MAEs) (He et al., 2022) instead learn by reconstructing masked patches and have been effective for pre-training encoders. However, standard MAE-style time-series encoders usually produce variable-length patch-token sequences, requiring downstream aggregation such as mean pooling or a `[CLS]` token. In contrast, TS-Fingerprint treats the compact latent code $F'$ as the primary representation. Unlike token-based forecasting models such as TimeXer (Wang et al., 2024c)

and GAFormer (Xiao et al., 2024), our fingerprint tokens are designed as a fixed-size bottleneck for medical time-series representation learning, combining reconstruction with redundancy reduction.

# 3. Proposed Method: Fingerprint Tokens

We propose **TS-Fingerprint**, a framework that reframes medical time-series representation learning from a heuristic reconstruction task to a formally constrained *Redundancy-Constrained Information Maximization* problem. In this section, we first formalize the learning objective using information-theoretic bounds (Section 3.1) and then describe the specific neural architecture designed to restrict the hypothesis class to satisfy these bounds (Section 3.2).

## 3.1. Main Theoretical Framework

**Problem Formulation.** Let $\mathcal{X} \subset \mathbb{R}^{T \times C}$ be the instance space of medical time series governed by an unknown distribution $\mathcal{D}$. We seek a representation mapping $E_\theta : \mathcal{X} \to \mathbb{R}^{k \times d}$ that compresses an instance $x$ into a set of $k$ disentangled latent tokens $F'$.

Standard autoencoders implicitly treat the latent space as a passive compression bucket, maximizing a lower bound on the mutual information $I(X; F')$ via reconstruction. However, we argue that *sufficiency* alone leads to entangled representations where physiological factors are diffused across dimensions. To address this, we formalize the representation learning problem as **Redundancy-Constrained Information Maximization**. We impose a strict independence constraint, seeking to maximize the *Information Bottleneck* capacity while simultaneously minimizing the *Total Correlation* (TC) of the latent space.

**Theorem 3.1** (Variational View of Redundancy-Constrained Information Maximization). *Let $F'$ be the latent representation and $X$ be the input. Minimizing the composite objective $\mathcal{L}_{total} = \mathcal{L}_{rec} + \lambda \mathcal{L}_{div}$ (defined in Sec 3.3) can be viewed as optimizing a variational surrogate of:*

$$\max_\theta \underbrace{I(X; F')}_{Sufficiency} - \lambda \underbrace{TC(F')}_{Independence} . \qquad (1)$$

*Specifically, $\mathcal{L}_{rec}$ maximizes the Evidence Lower Bound (ELBO) of the mutual information $I(X; F')$, while $\mathcal{L}_{div}$ minimizes the Total Correlation $TC(F') = \sum_{i=1}^{k} H(f'_i) - H(F')$ under the assumption that the latent distribution is Gaussian.*

*Proof.* See Supplemental Material A.1. The proof derives the Variational Lower Bound for the joint objective, demonstrating that $\mathcal{L}_{rec}$ maximizes the expected data log-likelihood (a lower bound on $I(X; F')$), while $\mathcal{L}_{div}$ minimizes the multi-information term, which aligns with Total

Correlation for Gaussian variables. □

## 3.2. Fingerprint Architecture: Restricting the Hypothesis Class

We propose the **Fingerprint Architecture** as the structural realization of Theorem 3.1. We reflect on the fact that standard Transformers induce a hypothesis space with excessive capacity, which encourages the memorization of high-frequency noise. To constrain the hypothesis class and enforce the theoretical bounds defined above, we *repurpose* the encoder-decoder structure.

**Encoder: The Duty of Disentangled Compression** To implement the capacity constraint of the Information Bottleneck, we propose our embedding process as follow: **1. Fixed-Rank Bottleneck:** Instead of maintaining the temporal sequence length, we define a learnable query set $Q \in \mathbb{R}^{k \times d}$ where $k \ll T$. The mapping $F' = \text{Attention}(Q, K = P, V = P)$ projects the high-dimensional input $P$ onto a fixed $k$-dimensional subspace; **2. Inductive Bias via Competition:** This architecture acts as a structural regularizer. By fixing the number of latent slots to $k$, we force the learned queries to compete for signal variance, discarding high-frequency variations that do not align with the principal physiological components defined by $Q$.

**Decoder: The Duty of Sufficiency** To ensure that the low-rank projection $F'$ remains a *sufficient statistic* for the input $X$ (preserving $I(X; F')$), we employ a Masked Reconstruction objective. Crucially, unlike standard MAE approaches that may leak information through visible patches, our decoder $D_\phi$ conditions generation *solely* on the Fingerprint tokens $F'$, where the mask tokens only contains the positional embedding of the original patches. This enforces a strict data processing inequality chain $X \to F' \to \hat{X}$, ensuring that no diagnostic information bypasses the bottleneck.

## 3.3. Learning Objectives: Geometry of the Latent Space

We formulate the training objective not merely as error minimization, but as a geometric regularization process that shapes the latent manifold to satisfy the bounds in Theorem 3.1. The total loss acts as a dual-force mechanism: an attractive force ensuring data fidelity and a repulsive force ensuring token disentanglement.

**Reconstruction Loss (Enforcing Sufficiency)** To strictly enforce the **Sufficiency** term ($I(X; F')$), we maximize the data log-likelihood. Under the standard assumption of Gaussian decoder noise, maximizing the variational lower bound is equivalent to minimizing the Mean Squared Error:

$$\mathcal{L}_{\text{rec}} = \mathbb{E}_{x \sim \mathcal{D}} \left[ \|x - D_\phi(T_{\text{mask}}, F')\|^2 \right] . \qquad (2)$$

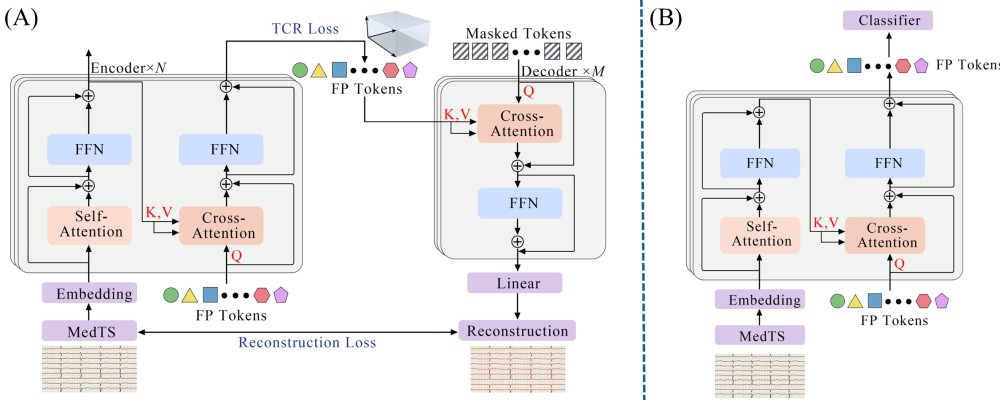

*Figure 2.* The TS-Fingerprint Framework. We achieve Redundancy-Constrained Information Maximization through a dual-stage process. (Left) Pre-Training: The Fingerprint Encoder ($E_\theta$) uses iterative cross-attention to compress variable-length MedTS patches into a fixed set of $k$ latent tokens. These tokens are shaped by two forces: (1) Disentanglement, where a Total Coding Rate (TCR) loss ($\mathcal{L}_{\text{div}}$) geometrically forces the tokens to be orthogonal; and (2) Sufficiency, where a Masked Decoder ($D_\phi$) reconstructs the signal strictly from the bottleneck tokens. (Right) Downstream Task: The pre-trained encoder encodes the MedTS into disentangled Fingerprint Tokens then fed into a classifier for clinical inference.

By conditioning the decoder strictly on $F'$, we force the bottleneck to retain the necessary statistics required to reconstruct the complex physiological morphology of the input.

**Diversity Loss (Maximizing Latent Volume)**   Minimizing reconstruction error alone does not guarantee a disentangled basis. For example, standard autoencoders often suffer from *feature collapse*, where multiple tokens degenerate into identical principal components. To enforce the **Independence** term (minimizing $\text{TC}(F')$), we treat latent tokens as vectors in a high-dimensional space and maximize the volume of the parallelotope they span. We employ the **Total Coding Rate (TCR)** (Yu et al., 2020) as a geometric regularizer:

$$\mathcal{L}_{\text{div}} = -\frac{1}{2} \log \det \left( I + \frac{d}{k\epsilon^2} F'^\top F' \right). \tag{3}$$

Here, we minimize the negative log-determinant (equivalently maximizing the volume). This acts as an explicit repulsive force between tokens.

**Lemma 3.2** (Orthogonality and Volume Maximization). *Let $\Sigma_{F'}$ be the empirical covariance matrix of the latent tokens. The Total Coding Rate is maximized (and redundancy minimized) if and only if $\Sigma_{F'}$ is diagonal. Minimizing $\mathcal{L}_{\text{div}}$ under a fixed energy constraint is equivalent to enforcing pairwise orthogonality $\langle f'_i, f'_j \rangle \approx 0$ for $i \neq j$.*

*Proof.* See Supplemental Material A.2. ☐

Geometrically, optimizing $\mathcal{L}_{\text{div}}$ under a fixed energy constraint (e.g., normalization of tokens) forces the tokens to become pairwise orthogonal, i.e., $\langle f'_i, f'_j \rangle \approx 0$ for $i \neq j$.

**Proposition 3.3** (Approximation Error and Mode Coverage). *Let $X \in \mathbb{R}^D$ be a centered random vector with covariance matrix $\Sigma_X$. Let $\{\lambda_j, v_j\}_{j=1}^D$ be the eigenvalues and eigenvectors of $\Sigma_X$, ordered such that $\lambda_1 \geq \cdots \geq \lambda_D \geq 0$. Let $m$ be the effective rank of the data manifold such that $\lambda_j \approx 0$ for $j > m$. A single-token bottleneck (such as single class token) $(k = 1)$ incurs a mean squared approximation error lower-bounded by $\sum_{j=2}^D \lambda_j$. For a $k$-token bottleneck $(k < m)$, explicitly penalizing token correlation via $\mathcal{L}_{\text{div}}$ (enforcing orthogonality) reduces the approximation error lower bound to $\sum_{j=k+1}^D \lambda_j$.*

*Proof.* See Supplemental Material A.3. ☐

*Remark* 3.4. This proposition provides the linear theoretical motivation for our Fingerprint view. While our implementation uses a non-linear deep decoder, this linear bound illustrates the fundamental geometric intuition: by ensuring the approximation error is bounded by the tail energy $\sum_{j=k+1}^D \lambda_j$, we minimize the risk of mode collapse. This geometric constraint biases the model to cover diverse spectral components rather than repeatedly encoding the dominant signal, potentially capturing lower-variance clinical features (like subtle arrhythmias).

### 3.4. Downstream Clinical Inference

We posit that a clinically useful representation must be not only sufficient but also **selective**. In medical diagnostics, pathology is often driven by sparse biomarkers (e.g., a specific arrhythmia segment) rather than global signal statistics. To model this, we formulate the downstream classification task via attention pooling mechanism that aggregates the

learned Fingerprint Tokens:

$$z = \sum_{i=1}^{k} \alpha_i f_i', \quad \text{where } \alpha_i = \text{Softmax}(\langle q_{\text{task}}, f_i' \rangle), \quad (4)$$

where $q_{\text{task}}$ are task specific linear layer. This formulation serves as a probe: if the representation $F'$ is truly disentangled, the attention weights $\alpha$ should naturally become sparse, selecting only the clinically relevant tokens. We formalize this intuition in Proposition 3.5, identifying the theoretical conditions under which this sparse recovery is feasible.

**Proposition 3.5** (Sample Complexity of Sparse Recovery). *Let the true labeling function $h^* : \mathbb{R}^k \rightarrow \mathcal{Y}$ depend on a subset of factors $S \subset \{1, \ldots, k\}$ with cardinality $|S| = s$. If the representation is entangled (modeled as a random rotation of the ground truth), learning $h^*$ requires a dense hypothesis class. The sample complexity scales linearly with the factor dimension: $m_{ent} \in \Omega(k)$. If the representation is factorized such that factors are axis-aligned with $S$, the optimal predictor lies in the class of $s$-sparse linear separators. The sample complexity scales logarithmically with the factor dimension: $m_{dis} \in O(s \log k)$. Consequently, for pathology detection driven by sparse biomarkers ($s \ll k$), the disentangled representation strictly improves learnability ($m_{dis} \ll m_{ent}$) in the low-data regime.*

*Proof.* See Supplemental Material A.4. $\qquad \square$

Beyond learnability, clinical deployment demands stability. Physiological signals are inherently noisy. Thus a trustworthy representation must distinguish between measurement noise and pathological shifts. We analyze this property by modeling the sensitivity of the predictor to local perturbations in the latent space. In Theorem 3.6, we establish a direct link between the geometry of the latent space (specifically, the orthogonality we enforce) and the worst-case robustness of the classifier.

**Theorem 3.6** (Robustness to Uncorrelated Noise). *Let $F \in \mathbb{R}^{d \times k}$ be a fixed latent representation consisting of $k$ standardized tokens ($\|f_i\|_2 = 1$) with Gram matrix $G = F^T F$. Consider a linear readout vector $w \in \mathbb{R}^d$ required to produce a target activation pattern $y \in \mathbb{R}^k$ on the tokens (i.e., satisfying the constraint $F^T w = y$).*

*For isotropic input noise $\eta \sim \mathcal{N}(0, \sigma^2 I_d)$, the Noise Sensitivity is $\mathcal{S} = \sigma^2 \|w\|_2^2$. The worst-case sensitivity over all unit-norm target patterns ($\|y\|_2 = 1$) using the minimum-norm readout $w$ is lower-bounded by the spectral properties of $G$:*

$$\max_{\|y\|=1} \mathcal{S}(w, F) \geq \frac{\sigma^2}{\lambda_{\min}(G)} \quad (5)$$

*Consequently, this worst-case robustness is strictly maximized if and only if the tokens are pairwise orthogonal ($G = I$).*

*Proof.* See Supplemental Material A.5. $\qquad \square$

## 4. Experiments

We evaluate TS-Fingerprint on seven real-world benchmarks spanning ECG, EEG, and human activity recognition. We assess its downstream classification performance, the effect of the redundancy-constrained pre-training objective, sensitivity to architectural choices, and whether the learned tokens exhibit measurable redundancy reduction and functional specialization.

**Baselines** We compare against general time-series backbones, including Autoformer (Wu et al., 2021), Crossformer (Zhang & Yan, 2023), FEDformer (Zhou et al., 2022), Informer (Zhou et al., 2021), MTST (Zhang et al., 2024), iTransformer (Liu et al., 2023), Nonformer (Liu et al., 2022), PatchTST (Nie, 2022), Reformer (Kitaev et al., 2020), and Transformer (Vaswani et al., 2017); the medical backbone Medformer (Wang et al., 2024b); and self-supervised baselines Ti-MAE (Li et al., 2023) and SimMTM (Dong et al., 2023). Additional foundation-style and medical SSL baselines are discussed in Section 4.6.

### 4.1. Evaluations on Representative Medical Datasets

We conduct classification experiments on seven popular real-world medical benchmarks and report 5 evaluation metrics: Accuracy, Precision, Recall, F1-score, and AUROC. We strictly split datasets using a subject-wise protocol to prevent data leakage. All experiments are repeated 5 times based on different seeds, and mean metrics are reported. All experiments are run on 2 NVIDIA RTX 3090 GPUs. For data augmentation, we adopt a frequency masking strategy during reconstruction (Zhang et al., 2022). We use a masking ratio of 0.6 and fingerprint number of 8, which provides a favorable balance between preserving physiologically meaningful spectral structure and introducing sufficient perturbation to reduce overfitting. Our architecture consists of a 6-layer encoder and a 2-layer decoder with hidden dimension of 128, both employing 8-head multi-head attention. We optimize the model using the Adam optimizer, with a learning rate of $1 \times 10^{-3}$ for pre-training and $1 \times 10^{-4}$ for downstream prediction tasks. We train for up to 100 epochs for both pre-training and downstream tasks, and apply early stopping based on validation F1 performance, with training halted if no improvement is observed for 10 consecutive epochs. The balancing coefficient $\lambda$ between the reconstruction loss and diversity loss is set to $1 \times 10^{-4}$.

Table 1 demonstrates the compelling performance of TS-

| Dataset | Metrics | Autoformer | Crossformer | FEDformer | Informer | iTransformer | MTST | Nonformer | PatchTST | Reformer | Transformer | Medformer | Ours |
|---|---|---|---|---|---|---|---|---|---|---|---|---|---|
| ADFTD (3-Classes) | Accuracy (%) | 45.25 ± 1.48 | 50.45 ± 2.31 | 46.30 ± 0.59 | 48.45 ± 1.96 | 52.60 ± 1.59 | 45.60 ± 2.03 | 49.95 ± 1.05 | 44.37 ± 0.95 | 50.78 ± 1.17 | 50.47 ± 2.14 | **53.27 ± 1.54** | **53.91 ± 0.54** |
| | Precision (%) | 43.67 ± 1.94 | 45.57 ± 1.63 | 46.05 ± 0.76 | 46.54 ± 1.68 | 46.79 ± 1.27 | 44.70 ± 1.33 | 47.71 ± 0.97 | 42.40 ± 1.13 | 49.64 ± 1.49 | 49.13 ± 1.83 | **51.02 ± 1.57** | **51.89 ± 0.49** |
| | Recall (%) | 42.96 ± 2.03 | 45.88 ± 1.82 | 44.22 ± 1.38 | 46.06 ± 1.84 | 47.28 ± 1.29 | 45.05 ± 1.30 | 47.46 ± 1.50 | 42.06 ± 1.48 | 49.89 ± 1.67 | 48.01 ± 1.53 | **50.71 ± 1.55** | **52.10 ± 1.09** |
| | F1-score | 42.59 ± 1.85 | 45.50 ± 1.70 | 43.91 ± 1.37 | 45.74 ± 1.38 | 46.79 ± 1.13 | 44.31 ± 1.74 | 46.96 ± 1.35 | 41.97 ± 1.37 | 47.94 ± 0.69 | 48.09 ± 1.59 | **50.65 ± 1.51** | **51.79 ± 0.74** |
| | AUROC | 61.02 ± 1.82 | 66.45 ± 2.03 | 62.62 ± 1.75 | 65.87 ± 1.27 | 67.26 ± 1.16 | 62.50 ± 0.81 | 66.23 ± 1.37 | 60.08 ± 1.50 | 69.17 ± 1.58 | 67.93 ± 1.59 | **70.93 ± 1.19** | **72.11 ± 0.73** |
| PTB (2-Classes) | Accuracy (%) | 73.35 ± 2.10 | 80.17 ± 3.79 | 76.05 ± 2.54 | 78.69 ± 1.68 | **83.89 ± 0.71** | 76.59 ± 1.90 | 78.66 ± 0.49 | 74.74 ± 1.62 | 77.96 ± 2.13 | 77.37 ± 1.02 | 83.50 ± 2.01 | **85.39 ± 0.63** |
| | Precision (%) | 72.11 ± 2.89 | 85.04 ± 4.83 | 77.58 ± 3.61 | 82.87 ± 1.02 | **88.25 ± 1.18** | 79.38 ± 1.90 | 82.77 ± 0.86 | 76.94 ± 1.51 | 81.72 ± 1.61 | 81.84 ± 0.66 | 85.19 ± 0.94 | **86.01 ± 0.72** |
| | Recall (%) | 63.24 ± 3.17 | 71.25 ± 6.29 | 66.10 ± 3.55 | 69.19 ± 2.90 | 76.39 ± 1.01 | 66.31 ± 2.95 | 69.12 ± 0.87 | 63.89 ± 2.71 | 68.20 ± 3.35 | 67.14 ± 1.80 | **77.11 ± 3.39** | **80.87 ± 0.91** |
| | F1-score | 63.69 ± 3.84 | 72.75 ± 7.19 | 67.14 ± 4.37 | 70.84 ± 3.47 | 79.06 ± 1.06 | 67.01 ± 1.32 | 69.74 ± 3.84 | 64.36 ± 3.38 | 69.65 ± 3.88 | 68.47 ± 2.19 | **79.18 ± 3.31** | **82.41 ± 1.11** |
| | AUROC | 78.54 ± 3.48 | 88.55 ± 3.45 | 85.93 ± 4.31 | 92.09 ± 0.53 | 91.18 ± 1.16 | 86.86 ± 2.75 | 89.37 ± 2.51 | 88.79 ± 0.91 | 91.13 ± 0.74 | 90.08 ± 1.76 | **92.81 ± 1.48** | **92.99 ± 0.98** |
| PTB-XL (5-Classes) | Accuracy (%) | 61.68 ± 2.72 | **73.30 ± 0.14** | 57.20 ± 9.47 | 71.43 ± 0.32 | 69.28 ± 0.22 | 72.14 ± 0.27 | 70.56 ± 0.55 | 73.23 ± 0.25 | 71.72 ± 0.43 | 70.59 ± 0.44 | 72.87 ± 0.23 | **73.98 ± 0.21** |
| | Precision (%) | 51.60 ± 1.64 | 65.06 ± 0.35 | 52.38 ± 6.09 | 62.64 ± 0.60 | 59.59 ± 0.45 | 63.84 ± 0.72 | 61.57 ± 0.65 | **65.70 ± 0.64** | 63.12 ± 1.02 | 61.57 ± 0.65 | 64.14 ± 0.42 | **66.45 ± 0.38** |
| | Recall (%) | 49.10 ± 1.52 | **61.23 ± 0.33** | 49.04 ± 7.26 | 59.12 ± 0.47 | 54.62 ± 0.18 | 60.01 ± 0.81 | 57.75 ± 0.72 | 60.82 ± 0.76 | 59.20 ± 0.75 | 57.62 ± 0.35 | 60.60 ± 0.46 | **62.21 ± 0.21** |
| | F1-score | 48.85 ± 2.27 | 62.59 ± 0.14 | 47.89 ± 8.44 | 60.44 ± 0.43 | 56.20 ± 0.19 | 61.43 ± 0.38 | 59.10 ± 0.66 | **62.61 ± 0.34** | 60.69 ± 0.18 | 59.05 ± 0.25 | 62.02 ± 0.37 | **63.51 ± 0.34** |
| | AUROC | 82.04 ± 1.44 | **90.02 ± 0.06** | 82.13 ± 4.17 | 88.65 ± 0.09 | 86.71 ± 0.10 | 88.97 ± 0.33 | 88.32 ± 0.36 | 89.74 ± 0.19 | 88.80 ± 0.24 | 88.21 ± 0.16 | 89.66 ± 0.13 | **90.41 ± 0.11** |
| APAVA (2-Classes) | Accuracy (%) | 68.64 ± 1.82 | 73.77 ± 1.95 | 74.94 ± 2.15 | 73.11 ± 4.40 | 74.55 ± 1.66 | 71.14 ± 1.59 | 71.89 ± 3.81 | 67.03 ± 1.65 | 78.70 ± 2.00 | 76.30 ± 4.72 | **78.74 ± 0.64** | **80.29 ± 0.32** |
| | Precision (%) | 68.48 ± 2.10 | 79.29 ± 4.36 | 74.59 ± 1.50 | 75.17 ± 6.06 | 74.77 ± 2.10 | 79.30 ± 0.97 | 78.76 ± 1.28 | **82.50 ± 3.95** | 77.64 ± 5.95 | 81.11 ± 0.84 | | **81.62 ± 0.44** |
| | Recall (%) | 68.77 ± 2.27 | 68.86 ± 1.70 | 73.56 ± 3.55 | 69.17 ± 4.56 | 71.76 ± 1.72 | 65.27 ± 2.17 | 69.44 ± 3.56 | 59.91 ± 2.02 | 75.00 ± 1.61 | 73.09 ± 5.01 | **75.40 ± 0.66** | **77.61 ± 0.49** |
| | F1-score | 68.06 ± 1.94 | 68.93 ± 1.85 | 73.51 ± 3.39 | 69.47 ± 5.06 | 72.30 ± 1.79 | 68.87 ± 2.34 | 70.55 ± 2.96 | 65.65 ± 2.08 | 73.94 ± 1.40 | 73.23 ± 2.67 | **76.31 ± 0.71** | **78.50 ± 0.31** |
| | AUROC | 75.94 ± 3.61 | 82.39 ± 3.52 | 83.72 ± 1.97 | 70.46 ± 4.91 | **85.59 ± 1.55** | 68.87 ± 2.34 | 70.55 ± 2.96 | 65.65 ± 2.08 | 73.94 ± 1.40 | 73.23 ± 2.67 | 83.20 ± 0.91 | **86.49 ± 1.00** |
| SleepEDF (5-Classes) | Accuracy (%) | 68.78 ± 1.23 | 80.76 ± 0.91 | 69.16 ± 1.14 | 73.60 ± 1.02 | 81.68 ± 0.74 | 82.40 ± 0.83 | 75.51 ± 0.96 | 81.35 ± 0.87 | 72.99 ± 1.18 | 70.30 ± 1.27 | **84.05 ± 0.76** | **84.18 ± 0.72** |
| | Precision (%) | 64.53 ± 1.34 | 71.73 ± 1.02 | 62.25 ± 1.29 | 61.83 ± 1.21 | 72.21 ± 0.88 | **73.52 ± 0.85** | 67.49 ± 1.03 | 71.89 ± 0.97 | 69.61 ± 1.11 | 60.87 ± 1.38 | 72.84 ± 0.79 | **74.55 ± 0.71** |
| | Recall (%) | 57.20 ± 1.41 | 72.27 ± 1.05 | 58.16 ± 1.33 | 61.59 ± 1.26 | 72.83 ± 0.86 | 73.30 ± 0.89 | 64.81 ± 1.08 | 70.36 ± 0.99 | 59.21 ± 1.30 | 61.14 ± 1.24 | **73.42 ± 0.82** | **74.36 ± 0.75** |
| | F1-score (%) | 59.42 ± 1.36 | 71.85 ± 1.01 | 59.10 ± 1.31 | 60.96 ± 1.22 | 72.36 ± 0.82 | **73.16 ± 0.86** | 63.55 ± 1.04 | 70.63 ± 0.93 | 61.92 ± 1.19 | 60.41 ± 1.28 | 71.49 ± 0.84 | **74.16 ± 0.70** |
| | AUROC (%) | 88.38 ± 0.94 | 94.21 ± 0.63 | 87.82 ± 0.97 | 89.14 ± 0.88 | 93.81 ± 0.54 | 94.56 ± 0.51 | 90.57 ± 0.79 | 94.42 ± 0.58 | 92.03 ± 0.71 | 87.82 ± 0.92 | **95.23 ± 0.52** | **95.37 ± 0.49** |
| FLAAP (10-Classes) | Accuracy (%) | 71.66 ± 1.43 | 75.84 ± 0.52 | 66.05 ± 1.37 | 73.72 ± 0.71 | 71.77 ± 2.25 | 71.58 ± 1.23 | **76.68 ± 1.46** | 56.46 ± 4.21 | 71.65 ± 1.27 | 74.96 ± 1.25 | 76.44 ± 0.64 | **77.59 ± 0.83** |
| | Precision (%) | 70.93 ± 1.54 | 74.99 ± 0.94 | 64.50 ± 1.03 | 72.10 ± 2.37 | 71.74 ± 1.54 | 70.90 ± 4.21 | **76.49 ± 3.81** | 51.14 ± 3.19 | 73.99 ± 1.02 | 74.22 ± 1.57 | 75.76 ± 0.75 | **76.83 ± 0.77** |
| | Recall (%) | 71.12 ± 2.19 | 74.49 ± 1.17 | 65.26 ± 2.24 | 70.24 ± 0.56 | 70.51 ± 0.73 | 69.94 ± 2.38 | **76.74 ± 1.49** | 51.74 ± 2.43 | 73.83 ± 0.85 | 73.97 ± 0.97 | 75.10 ± 0.81 | **76.76 ± 1.02** |
| | F1-score | 73.22 ± 1.89 | 75.64 ± 0.66 | 63.59 ± 1.70 | 72.32 ± 1.69 | 70.96 ± 0.91 | 71.15 ± 0.94 | 76.14 ± 0.89 | 53.98 ± 2.82 | 71.14 ± 1.45 | 74.49 ± 1.39 | **76.25 ± 0.65** | **77.48 ± 0.89** |
| | AUROC | 96.17 ± 0.97 | 96.81 ± 0.44 | 93.20 ± 2.65 | 96.25 ± 0.57 | 95.96 ± 1.49 | 95.42 ± 1.95 | **96.27 ± 0.86** | 88.96 ± 1.77 | 96.91 ± 0.47 | 94.40 ± 1.26 | 95.84 ± 1.15 | **96.76 ± 0.47** |
| UCI-HAR (6-Classes) | Accuracy (%) | 82.38 ± 2.31 | 89.74 ± 1.08 | 90.16 ± 0.81 | 90.30 ± 0.36 | 84.30 ± 1.15 | 89.79 ± 0.31 | 90.01 ± 0.47 | 87.11 ± 1.28 | 88.44 ± 2.02 | 88.86 ± 1.65 | **91.65 ± 0.74** | **90.87 ± 1.21** |
| | Precision (%) | 83.94 ± 1.89 | **90.96 ± 0.76** | 89.36 ± 0.82 | 90.26 ± 0.53 | 84.38 ± 2.21 | 89.58 ± 0.48 | 90.19 ± 0.37 | 87.46 ± 2.17 | 88.69 ± 1.87 | 89.03 ± 0.72 | **91.89 ± 0.59** | 90.61 ± 1.01 |
| | Recall (%) | 83.29 ± 1.74 | 89.32 ± 0.95 | 90.50 ± 0.58 | 90.31 ± 0.66 | 84.25 ± 0.97 | 89.22 ± 0.66 | 90.14 ± 0.52 | 87.05 ± 0.88 | 88.44 ± 2.02 | 88.86 ± 1.65 | **91.65 ± 0.74** | **90.87 ± 1.21** |
| | F1-score | 80.82 ± 2.04 | 89.70 ± 1.10 | 90.43 ± 1.02 | 90.21 ± 0.79 | 84.28 ± 0.76 | 89.31 ± 0.29 | 90.87 ± 0.19 | 88.88 ± 1.26 | 88.41 ± 1.98 | 88.80 ± 1.67 | **91.61 ± 0.75** | **90.92 ± 1.19** |
| | AUROC | 94.21 ± 1.14 | 98.20 ± 0.41 | 98.36 ± 0.57 | 98.48 ± 0.69 | 96.55 ± 1.02 | 98.35 ± 0.37 | 97.97 ± 0.52 | 98.35 ± 0.47 | 98.21 ± 0.57 | 98.08 ± 0.88 | **98.99 ± 0.34** | **98.67 ± 0.79** |

*Table 1.* **Main results on mainstream medical classification benchmarks.** TS-Fingerprint achieves the **top-1 average rank** across 5 metrics, surpassing extensive state-of-the-art baselines including medical specialists (e.g., Medformer). Notably, TS-Fingerprint surpasses specialized architectures without requiring complex multi-granularity engineering, validating the superiority of our information-theoretic bottleneck. Red: best results. Blue: second best results.

Fingerprint. Concretely, our method achieves the top-1 average rank(average rank:1.24) across all metrics, surpassing 11 baselines. We evaluate our model against the second-best baseline, Medformer, using a one-tailed Wilcoxon signed-rank test. All results are statistically significant ($p < 0.05$), demonstrating the superiority of our proposed model. It is notable that general-purpose models like PatchTST struggle on complex EEG tasks (ADFTD). We attribute this failure to their Entangled View. By prioritizing global MSE, these architectures smooth over subtle, high-frequency neurodegenerative markers in favor of dominant amplitudes, resulting in insufficient representation of pathology. Furthermore, TS-Fingerprint outperforms specialized architectures like Medformer without requiring complex multi-granularity engineering. This suggests that our Fingerprint View, which forces compression into orthogonal tokens, is inherently more effective at isolating signal from noise than hand-crafted hierarchies and elegant in design.

### 4.2. Validating the Redundancy-Constrained Objective

To verify the necessity of our proposed self-supervised pre-train strategy, we conduct a controlled ablation study on ADFTD and PTB-XL. We compare three configurations: Scratch, where the encoder is trained solely via supervised cross-entropy from random initialization, Pretrained($\mathcal{L}_{rec}$) and Pretrained($\mathcal{L}_{rec} + \mathcal{L}_{div}$), where the model is first optimized using different configurations of loss functions. All other hyperparameters are kept identical to ensure a fair comparison. Table 2 demonstrates the effectiveness of our design. As expected, the pre-training stage brings a performance promotion on both datasets. Notably, on the challenging ADFTD benchmark, pre-training yields a remarkable +13.1% relative improvement in F1-score. This indicates that our redundancy-constrained objective successfully initializes the latent space with structured, population-level priors, which is crucial for preventing overfitting in scenarios with high inter-subject heterogeneity. On the larger and more diverse PTB-XL cohort, we observe consistent gains (+8.5% F1), confirming that our method scales effectively and provides a substantially better starting point than random initialization.

| Dataset | Init. | Acc. (%) | F1 | AUROC |
|---|---|---|---|---|
| ADFTD | Scratch | 48.83 | 45.80 | 65.29 |
| | Pre-trained ($\mathcal{L}_{rec}$) | 53.51 | 50.48 | 70.21 |
| | Pre-trained ($\mathcal{L}_{rec} + \mathcal{L}_{div}$) | 53.91 (+Δ10.4%) | 51.79 (+Δ13.1%) | 72.11 (+Δ10.5%) |
| PTB-XL | Scratch | 71.29 | 58.52 | 86.73 |
| | Pre-trained ($\mathcal{L}_{rec}$) | 73.56 | 62.66 | 90.11 |
| | Pre-trained ($\mathcal{L}_{rec} + \mathcal{L}_{div}$) | 73.98 (+Δ3.8%) | 63.51 (+Δ8.5%) | 90.41 (+Δ4.2%) |

*Table 2.* **Ablation of the Redundancy-Constrained Pre-training strategy.** Δ indicate the percentage of performance gain compared with results without pretrain. We compare the performance of TS-Fingerprint trained from **Scratch** (random initialization) versus **Pre-trained** (using only reconstruction objective) and **Pre-trained** (using our reconstruction + diversity objective). Pre-training yields a consistent performance promotion across both datasets.

To verify architectural robustness, we analyze the impact of the number of fingerprint tokens ($k \in \{6, 8, 10\}$) and the masking ratio ($r \in \{0.5, \dots, 0.8\}$) based on PTB-XL.

These parameters jointly determine the information bottleneck size and the difficulty of the self-supervised reconstruction task. Table 3 demonstrates that TS-Fingerprint is highly robust to hyperparameter variations. Concretely, performance peaks at $k = 8$. Smaller bottlenecks limit the capacity to capture physiological heterogeneity, while larger sets ($k = 10$) introduce redundancy that weakens the disentanglement effect. Regarding the mask ratio, the model remains stable across $0.5 \sim 0.7$ but degrades at 0.8. This indicates that while aggressive masking fosters global context learning, removing excessive signal eliminates the local cues necessary for fine-grained diagnosis. Consequently, TS-Fingerprint offers a wide effective operating region ($k = 8, r = 0.6$) without requiring extensive tuning.

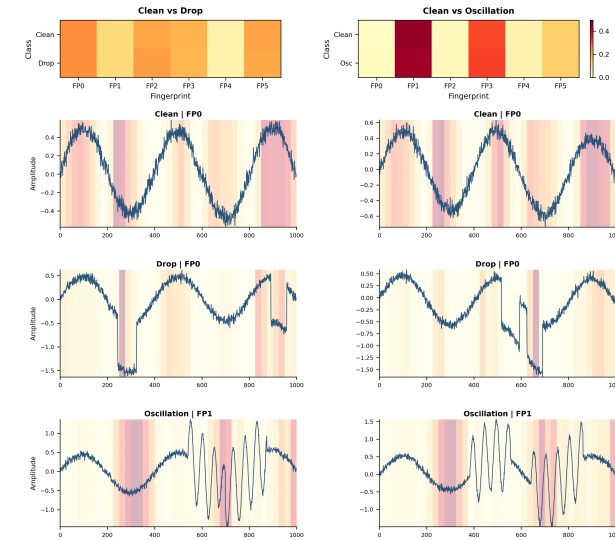

*Figure 3.* **Mechanistic disentanglement analysis via controlled perturbations. Top:** Average attention allocation of fingerprint tokens (`FP0--FP5`) under three signal regimes. The sparse activation patterns indicate that different tokens specialize in distinct signal characteristics. **Bottom:** Token-specific attention heatmaps overlaid on raw signals. Crucially, the model exhibits a clear division of labor: `FP0` sharply aligns with amplitude drops, while `FP1` selectively tracks oscillatory bursts. This confirms that TS-Fingerprint effectively disentangles independent factors of variation (local anomalies) from the global signal context, preventing the representational collapse common in holistic models.

| FP | Dim | Mask Rate | Acc | Precision | Recall | F1 | AUROC |
|---|---|---|---|---|---|---|---|
| | | | FP=6 | | | | |
| 6 | 256 | 0.5 | 0.7019 | 0.6263 | 0.5377 | 0.5481 | 0.8727 |
| 6 | 256 | 0.6 | 0.7071 | 0.6312 | 0.5469 | 0.5668 | 0.8861 |
| 6 | 256 | 0.7 | 0.7063 | 0.6266 | 0.5411 | 0.5541 | 0.8753 |
| 6 | 256 | 0.8 | 0.6977 | 0.6163 | 0.5321 | 0.5432 | 0.8708 |
| | | | FP=8 | | | | |
| 8 | 256 | 0.5 | **0.7407** | **0.6654** | 0.6102 | 0.6260 | 0.9025 |
| 8 | 256 | 0.6 | 0.7398 | 0.6645 | **0.6221** | **0.6351** | **0.9041** |
| 8 | 256 | 0.7 | 0.7386 | 0.6635 | 0.6050 | 0.6210 | 0.9004 |
| 8 | 256 | 0.8 | 0.7235 | 0.6591 | 0.5750 | 0.5919 | 0.8904 |
| | | | FP=10 | | | | |
| 10 | 256 | 0.5 | 0.6736 | 0.5836 | 0.5053 | 0.5176 | 0.8564 |
| 10 | 256 | 0.6 | 0.6641 | 0.5532 | 0.4991 | 0.5070 | 0.8450 |
| 10 | 256 | 0.7 | 0.7107 | 0.6488 | 0.5524 | 0.5633 | 0.8785 |
| 10 | 256 | 0.8 | 0.7042 | 0.6411 | 0.5440 | 0.5570 | 0.8742 |

*Table 3.* **Sensitivity analysis on core hyperparameters.** We observe that $K = 8$ offers an optimal trade-off for the fingerprint token count. Smaller $K$ creates an overly constrictive bottleneck, while $K = 10$ weakens the division of labor through redundancy. For the mask ratio ($r$), the model maintains stability until $r \geq 0.8$, where performance declines due to the loss of fine-grained local cues necessary for diagnosis.

### 4.3. Mechanistic Validation of Disentanglement via Controlled Perturbations

Standard statistical metrics often fail to reveal how latent factors respond to specific physiological anomalies. To bridge this gap, we construct a controlled synthetic benchmark containing three distinct signal regimes: (i) Amplitude Drops (simulating sensor failure), (ii) Oscillatory Bursts (simulating transient high-frequency activity), and (iii) Clean Baselines. We analyze the functional specialization of our model by visualizing the class-wise attention allocation of fingerprint tokens under these perturbations. More details about the synthetic dataset are in Appendix D.

Figure 3 provides direct mechanistic evidence of disentanglement. As shown in the heatmaps, TS-Fingerprint exhibits a sparse and systematic division of labor. Concretely, **FP0** sharply aligns with amplitude discontinuities, while **FP1** selectively tracks oscillatory bursts. This behavior stands in sharp contrast to holistic representations, which typically

diffuse perturbation information across all latent dimensions (an Entangled View).

Crucially, this token-level specialization emerges naturally from our redundancy-constrained objective without explicit supervision. While real-world physiological signals involve more complex interactions, this controlled experiment validates the functional capacity of our framework. It demonstrates that the orthogonality constraint is effectively driving the model to isolate independent factors of variation (like local anomalies) rather than memorizing entangled noise, providing a mechanistic basis for the improved robustness observed in clinical benchmarks.

### 4.4. Interpretability Analysis: Adaptive Sparsity

To explore how our fingerprint tokens potentially adapt to different physiological complexities, we visualize the class-wise attention distribution over the 8 fingerprint slots for both ECG (PTB-XL) and EEG (ADFTD) tasks. This allows us to probe whether the model learns rigid features or adapts its utilization of the latent space based on signal entropy.

Figure 4 reveals a distinct Adaptive Sparsity mechanism. For ECG, the attention distribution is highly concentrated, with nearly all probability mass assigned to three dominant

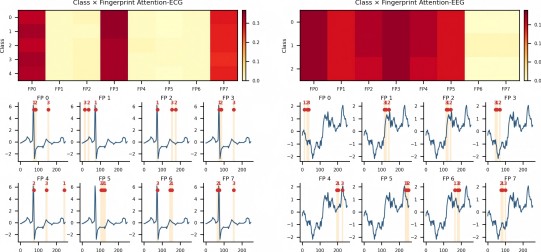

*Figure 4.* **Class-wise attention allocation across fingerprint slots for two physiological modalities.** Orange shading marks the top-$k$ attention windows, while red markers denote attention ranks. **Left:** ECG (PTB-XL) exhibits sharp, localized attention, where the model concentrates diagnostic information into a small subset of dominant slots. **Right:** EEG (ADFTD) displays distributed (but not uniform), multi-slot utilization, reflecting the higher entropy and complexity of cortical dynamics. These results demonstrate that TS-Fingerprint possesses an intrinsic Adaptive Sparsity mechanism, automatically tuning its usage of the latent bottleneck to match the structural complexity of the input signal without manual intervention.

slots (e.g., $k = 0, 3, 7$). This indicates that for structurally consistent signals like ECG, the model automatically isolates core morphological cues (such as QRS complexes) into a compact subset of tokens, leaving others dormant to reduce noise. In contrast, EEG signals trigger a substantially broader allocation across the latent space. This distributed pattern reflects the heterogeneous spatial-temporal dynamics of neurodegeneration, which requires integrating information across multiple latent factors.

These findings confirm that TS-Fingerprint avoids the one-size-fits-all compression typical of Fixed-Grid models (e.g., those relying on a single class token or global pooling). Instead, it dynamically allocates representational capacity in proportion to the intrinsic complexity of the data, achieving an optimal balance between interpretability (sparsity) and expressivity (distributedness).

### 4.5. Comparison with Ti-MAE and SimMTM

To quantify the transferability of the learned representations and disentangle the true contribution of pre-training strategies from the generic capacity of the backbone architecture, we benchmark TS-Fingerprint against Ti-MAE (Li et al., 2023) and SimMTM(Dong et al., 2023) on the heterogeneous ADFTD dataset. Our evaluation specifically focuses on Performance Promotion, which compares the relative F1 gain obtained when transitioning from training from scratch to fine-tuning on a pretrained model. This metric serves as a proxy for representational quality: a well-structured latent space should facilitate rapid and significant adaptation to downstream tasks.

Table 4 reveals a fundamental limitation in existing MAE frameworks. Ti-MAE exhibits almost no performance gain

($+0.3\%$) when unfrozen. This suggests that the baseline performance of such frameworks stems primarily from the generic capacity of their heavy backbones rather than the efficacy of the pre-training strategy. Because the learned latent space is entangled and rigid, the model essentially memorizes noise during pre-training, creating a representation that is structurally incompatible with discriminative tasks. While SimMTM shows better adaptation due to its invariant learning objectives, it still lags behind our method, indicating that holistic embeddings struggle to isolate specific pathological markers from background activity.

In contrast, TS-Fingerprint achieves a massive $13.1\%$ promotion under fine-tuning, which demonstrates that our Redundancy-Constrained objective does not just compress data; it factorizes it. By initializing the latent space with orthogonal, disentangled slots, our method provides a structure that is inherently amenable to specialization, allowing the fine-tuning stage to simply align existing factors to disease classes rather than untangling a chaotic global vector.

| Method | Setting | Accuracy (%) | F1 Score |
|---|---|---|---|
| Ti-MAE | Scratch | 50.27 | 45.88 |
| | Pre-trained | 50.48 (+$\Delta0.41\%$) | 45.44 (-$\Delta0.96\%$) |
| SimMTM | Scratch | 51.98 | 43.66 |
| | Pre-trained | 52.92(+$\Delta1.81\%$) | 45.59(+$\Delta4.42\%$) |
| Ours | Scratch | 48.83 | 45.80 |
| | Pre-trained | **53.91**(+$\Delta10.40\%$) | **51.79**(+$\Delta13.07\%$) |

*Table 4.* **Fine-tuning performance comparison on the ADFTD dataset.** We evaluate the transferability of representations by comparing the gain from pre-training across different self-supervised frameworks. While Ti-MAE and SimMTM show limited or negligible gains (indicating rigid, entangled latent spaces), **TS-Fingerprint achieves a substantial performance promotion** (e.g., boosting F1 from 45.80% to 51.79%).

### 4.6. Comparison with Foundation-Style and Medical SSL Baselines

To broaden the baseline coverage, we further compare TS-Fingerprint with recent foundation-style time-series models and modality-specific SSL methods on ADFTD. We include GPT4TS (Zhou et al., 2023) and TimesFM (Das et al., 2023) as representative large-scale time-series models, and CLOCS (Kiyasseh et al., 2021) and MTS-LOF (Li et al., 2024) as medical time-series SSL baselines. All methods are evaluated under the same ADFTD protocol. As shown in Table 5, TS-Fingerprint achieves the best F1 and AUROC among these additional baselines, while GPT4TS obtains slightly higher accuracy. The stronger F1 and AUROC suggest that the proposed fingerprint bottleneck is particularly effective under the class-imbalanced and heterogeneous ADFTD setting, where balanced discrimination is more informative than accuracy alone. These results complement the main comparison in Table 1 and indicate

| Method | Accuracy | Precision | Recall | F1 | AUROC |
|---|---|---|---|---|---|
| GPT4TS (Zhou et al., 2023) | 54.11±0.57 | 47.76±0.94 | 46.63±0.87 | 44.56±0.58 | 67.85±0.52 |
| TimesFM (Das et al., 2023) | 52.77±1.71 | 46.89±1.20 | 46.34±0.53 | 45.25±1.05 | 66.17±0.99 |
| CLOCS (Kiyasseh et al., 2021) | 49.31±0.88 | 46.86±0.74 | 43.67±0.81 | 45.21±0.90 | 67.31±0.84 |
| MTS-LOF (Li et al., 2024) | 52.07±0.77 | 45.63±0.71 | 50.52±0.93 | 47.95±0.82 | 68.89±0.77 |
| Ours | 53.91±0.54 | **51.89±0.49** | **52.10±1.09** | **51.79±0.74** | **72.11±0.73** |

*Table 5.* Comparison with foundation-style time-series models and medical SSL baselines on ADFTD.

| Aggregation Strategy | Accuracy (%) | F1 Score |
|---|---|---|
| Mean Pooling | 51.67 | 48.83 |
| Flattening | 53.63 | 49.83 |
| Attention Pooling | **53.91** | **51.79** |

*Table 6.* Ablation of downstream aggregation strategies on ADFTD. Attention pooling achieves the best F1 score, supporting the sparse-readout intuition in Proposition 3.5.

that the gains are not limited to conventional Transformer backbones.

### 4.7. Effect of Downstream Aggregation

Proposition 3.5 motivates a selective readout when labels depend on only a subset of latent factors. We therefore compare three aggregation strategies on ADFTD using the same pre-trained encoder and training protocol: mean pooling, which uniformly averages all fingerprint tokens; flattening, which applies a dense classifier to the concatenated tokens; and attention pooling, which adaptively weights tokens through the task-query mechanism in Eq. 4. As shown in Table 6, attention pooling performs best, improving F1 by 2.96 points over mean pooling and 1.96 points over flattening. This suggests that uniform aggregation can dilute task-relevant tokens, while a dense readout may not optimally exploit the bottleneck structure. The result is consistent with the sparse-readout intuition in Proposition 3.5.

### 4.8. Comparison of Diversity Regularizers

To assess the role of the diversity objective, we compare our log-determinant loss with two decorrelation regularizers, Barlow Twins (Zbontar et al., 2021) and the VICReg covariance loss (Bardes et al., 2021). All variants use the same architecture and training protocol; only the diversity

| Regularizer | Off-diagonal Corr. ↓ | Accuracy (%) ↑ | F1 Score ↑ |
|---|---|---|---|
| No constraint | 0.857 | 53.51 | 50.48 |
| Barlow Twins | 0.783 | 49.50 | 44.62 |
| VICReg covariance | 0.637 | 49.87 | 46.47 |
| Log-det diversity (Ours) | **0.022** | **53.91** | **51.79** |

*Table 7.* Comparison of diversity regularizers on ADFTD. All methods use the same fingerprint architecture and training protocol; only the diversity term is changed during pre-training. The proposed log-determinant loss achieves the lowest token correlation and the best downstream F1 score.

term is changed during pre-training. We also include a reconstruction-only baseline. We report ADFTD classification performance and the average off-diagonal token correlation, where lower values indicate less token redundancy. Table 7 shows that generic decorrelation losses do not necessarily improve the fingerprint representation. Although Barlow Twins and VICReg reduce token correlation relative to reconstruction-only pre-training, they reduce downstream performance in this setting. This may reflect a mismatch between their original use on embedding dimensions or paired views and our objective of diversifying a small set of reconstruction-preserving latent tokens. In contrast, the log-determinant loss yields the lowest token correlation while maintaining the best F1 score, consistent with the volume-maximization intuition in Lemma 3.2. We therefore use it as the default redundancy-reduction objective.

### 4.9. Effect of Multi-Scale Input Embedding

We also test whether TS-Fingerprint benefits from a stronger input tokenizer. Replacing the default linear patch embedding with a Medformer-style multi-scale embedding, while keeping the fingerprint encoder and objectives unchanged, improves ADFTD accuracy from 53.91 to 55.06 and F1 from 51.79 to 53.01. This result indicates that the fingerprint bottleneck is compatible with richer temporal tokenizers. We keep the vanilla embedding as the default to isolate the contribution of the bottleneck and leave systematic integration of hierarchical embeddings to future work.

## 5. Conclusion

This paper addresses the open question of how to learn interpretable, disentangled representations for medical time series without sacrificing performance. To this end, we propose TS-Fingerprint, a theoretically grounded framework that compresses physiological dynamics into a compact set of orthogonal tokens. By formulating the learning objective as a Redundancy-Constrained Information Maximization problem, we successfully bridge the gap between opaque deep learning models and the clinical need for verifiable digital biomarkers. Extensive experiments on five mainstream medical benchmarks demonstrate the superiority of our approach. As a disentangled bottleneck model, TS-Fingerprint achieves consistent state-of-the-art performance, surpassing both holistic Transformers and specialized medical backbones. Crucially, our analysis reveals that this performance stems from a clear division of labor: the model is capable of isolating independent modes of variation (e.g., specific arrhythmia markers vs. noise) into distinct slots. This confirms that enforcing disentanglement is not just an interpretability constraint, but a fundamental catalyst for robust generalization in clinical settings.

# Impact Statement

This work aims to advance the field of medical time-series representation learning by introducing a theoretically grounded, redundancy-constrained self-supervised framework. By enforcing structured disentanglement in the latent space, the proposed method enables more interpretable, sample-efficient, and robust representations of physiological signals such as ECG and EEG.

## Potential Positive Impact

The primary positive impact of this work lies in its contribution to foundational machine learning methodology for healthcare data. By producing compact and disentangled latent tokens that correspond to independent physiological factors, our approach facilitates improved generalization in low-data regimes and provides a more transparent interface for downstream clinical modeling. This may support the development of more reliable digital biomarkers, improve robustness across heterogeneous patient populations, and enhance model interpretability, an essential requirement for high-stakes medical decision support systems. Importantly, our framework is designed as a representation-learning tool rather than a diagnostic system, and is intended to assist researchers and clinicians in model development rather than replace expert judgment.

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

# A. Proofs of Theoretical Claims

## A.1. Proof of Theorem 3.1

*Proof.* We analyze the two terms of the objective function $\mathcal{L}_{\text{total}}$ independently, establishing their information-theoretic duals.

Recall the definition of Mutual Information: $I(X; F') = H(X) - H(X|F')$. Since the data distribution is fixed, $H(X)$ is constant $(C)$. Maximizing $I(X; F')$ is therefore equivalent to minimizing the conditional entropy $H(X|F')$.

We invoke the variational lower bound on mutual information (Barber & Agakov, 2004). For any variational distribution $p_\theta(X|F')$ (the decoder), the following inequality holds:

$$I(X; F') \geq H(X) + \mathbb{E}_{x \sim p(X), f' \sim q_\phi(F'|x)}[\log p_\theta(x|f')]. \quad (6)$$

Assume the decoder models the data as a Gaussian distribution with fixed scalar variance, $p_\theta(X|F') = \mathcal{N}(D_\theta(F'), \sigma^2 I)$. The log-likelihood is:

$$\log p_\theta(x|f') \propto -\frac{1}{2\sigma^2}|x - D_\theta(f')|^2 + \text{const.} \quad (7)$$

Negating the objective to frame this as a minimization problem, we obtain:

$$\min_{\theta, \phi} \mathbb{E}_{x, f'}\left[|x - D_\theta(f')|^2\right] \iff \max_{\theta, \phi} \mathbb{E}_{x, f'}[\log p_\theta(x|f')]. \quad (8)$$

Thus, minimizing the Mean Squared Error ($\mathcal{L}_{\text{rec}}$) maximizes the variational lower bound $\mathcal{I}_{LB}(X; F')$, enforcing sufficiency of the representation $F'$ for $X$.

The term $\mathcal{L}_{\text{div}}$ is defined via the Total Coding Rate (TCR). The Total Correlation (TC) for the random vector $F'$ is defined as the Kullback-Leibler divergence between the joint distribution and the product of marginals:

$$\text{TC}(F') = D_{KL}\left(p(F') \,\|\, \prod_{i=1}^{k} p(f'i)\right) \quad (9)$$

$$= \sum_{i=1}^{k} H(f'i) - H(F'). \quad (10)$$

Under the assumption that $F'$ follows a Gaussian distribution (consistent with the TCR formulation), the entropy is given by $H(F') = \frac{1}{2}\log\det(\Sigma_{F'}) + \text{const}$, where $\Sigma_{F'}$ is the covariance matrix of $F'$. The Total Correlation becomes:

$$\text{TC}(F') = \frac{1}{2}\sum i = 1^k \log(\Sigma_{ii}) - \frac{1}{2}\log\det(\Sigma_{F'}), \quad (11)$$

which is geometrically equivalent to the expansion vs. compression objective in $\mathcal{L}_{\text{div}}$ (Yu et al., 2020). Thus, minimizing $\mathcal{L}_{\text{div}}$ directly minimizes $\text{TC}(F')$, thereby enforcing statistical independence among the latent dimensions.

Therefore, minimizing $\mathcal{L}_{\text{total}} = \mathcal{L}_{\text{rec}} + \lambda \mathcal{L}_{\text{div}}$ corresponds to the constrained optimization problem:

$$\min_{\theta, \phi} \left(-\mathbb{E}[\log p_\theta(X|F')] + \lambda \text{TC}(F')\right). \quad (12)$$

This is formally equivalent to maximizing the variational information bound subject to an independence constraint:

$$\max_{\theta, \phi} \underbrace{\mathcal{I}_{LB}(X; F')}_{\text{Sufficiency}} - \lambda \underbrace{\text{TC}(F')}_{\text{Independence}}, \quad (13)$$

which concludes the proof. $\square$

## A.2. Proof of Lemma 3.2

*Proof.* We proceed by establishing the information-theoretic definition of redundancy and applying matrix inequalities to link this definition to the geometry of the covariance matrix.

Recall from Theorem 3.1 that redundancy is quantified by the Total Correlation:

$$\text{TC}(F') = \sum_{i=1}^{k} H(f'i) - H(F'). \quad (14)$$

Assuming $F'$ follows a multivariate Gaussian distribution $\mathcal{N}(\mu, \Sigma_{F'})$ (after normalization), the entropies are given by:

$$H(f'_i) = \frac{1}{2}\log(2\pi e \Sigma_0 ii) \quad (15)$$

$$H(F') = \frac{1}{2}\log(\det(2\pi e \Sigma_{F'}))$$
$$= \frac{1}{2}\log(\det(\Sigma_{F'})) + \text{const.} \quad (16)$$

Substituting these into the expression for $\text{TC}(F')$:

$$\text{TC}(F') = \frac{1}{2}\left(\sum_{i=1}^{k} \log(\Sigma_{ii}) - \log(\det(\Sigma_{F'}))\right)$$
$$= \frac{1}{2}\log\left(\frac{\prod_{i=1}^{k}\Sigma_{ii}}{\det(\Sigma_{F'})}\right). \quad (17)$$

We invoke **Hadamard's Inequality**, which states that for any positive semi-definite matrix $\Sigma_{F'}$:

$$\det(\Sigma_{F'}) \leq \prod_{i=1}^{k}\Sigma_{ii}. \quad (18)$$

Equality holds if and only if $\Sigma_{F'}$ is diagonal. Consequently, the term inside the logarithm is always $\geq 1$, and $\text{TC}(F') \geq 0$. The global minimum $\text{TC}(F') = 0$ is achieved if and only if equality holds in Hadamard's inequality, i.e., when $\Sigma_{F'}$ is diagonal.

The entries of the covariance matrix $\Sigma_{F'}$ are defined as $\Sigma_{ij} = \text{Cov}(f'_i, f'_j)$. If the data is centered (zero mean), this equates to the expected inner product:

$$\Sigma_{ij} = \mathbb{E}[\langle f'i, f'j \rangle]. \qquad (19)$$

A diagonal matrix implies that all off-diagonal entries are zero:

$$\Sigma_{F'} \text{ is diagonal } \iff \forall i \neq j, \qquad (20)$$
$$\Sigma_{ij} = 0 \implies \mathbb{E}[\langle f'_i, f'_j \rangle] = 0. \qquad (21)$$

Thus, minimizing redundancy implies decorrelation (orthogonality in expectation).

Consider the optimization problem where the "energy" (variance) of each feature is fixed, i.e., $\Sigma_{ii} = c$ for all $i$. The objective function simplifies to:

$$\min_{\Sigma_{F'}} \text{TC}(F') \iff$$
$$\min_{\Sigma_{F'}} (C - \log \det(\Sigma_{F'})) \iff$$
$$\max_{\Sigma_{F'}} \det(\Sigma_{F'}). \qquad (22)$$

Geometrically, $\sqrt{\det(\Sigma_{F'})}$ represents the volume of the parallelotope spanned by the covariance eigenvectors. Maximizing this volume subject to fixed edge lengths (fixed variances) forces the vectors to be orthogonal.

Therefore, minimizing $\mathcal{L}_{\text{div}}$ (which acts as a proxy for TC) under fixed energy constraints forces $\Sigma_{F'}$ to be diagonal, which is equivalent to enforcing pairwise orthogonality $\langle f'_i, f'_j \rangle \approx 0$.

$\square$

### A.3. Proof of Proposition 3.3

*Proof.* We analyze the reconstruction error $\mathcal{E} = \mathbb{E}[\|X - \hat{X}\|^2]$ under the assumption of a linear decoder (or a local linear approximation of the manifold), which allows for spectral analysis. The reconstruction is given by the orthogonal projection of $X$ onto the subspace spanned by the encoder weights $W \in \mathbb{R}^{D \times k}$.

We first do the decomposition of reconstruction error. Let $\mathcal{S}$ be the subspace spanned by the $k$ latent tokens (basis vectors $w_1, \ldots, w_k$). The optimal reconstruction $\hat{X}$ is the orthogonal projection of $X$ onto $\mathcal{S}$. The expected squared error is the trace of the residual covariance:

$$\mathcal{E}(\mathcal{S}) = \text{Tr}(\Sigma_X) - \text{Tr}(P_{\mathcal{S}} \Sigma_X P_{\mathcal{S}}), \qquad (23)$$

where $P_{\mathcal{S}}$ is the projection operator onto $\mathcal{S}$. Using the spectral decomposition $\Sigma_X = \sum_{j=1}^{D} \lambda_j v_j v_j^T$, the total variance is $\text{Tr}(\Sigma_X) = \sum_{j=1}^{D} \lambda_j$. Minimizing error is equivalent to maximizing the captured variance $\text{Tr}(P_{\mathcal{S}} \Sigma_X P_{\mathcal{S}})$.

In the single token case, for $k = 1$, the subspace $\mathcal{S}$ is spanned by a single unit vector $w_1$. The captured variance is maximized when $w_1$ aligns with the principal eigenvector $v_1$ (by the Rayleigh Quotient definition). The captured variance is $\lambda_1$. The approximation error is:

$$\mathcal{E}_{k=1} = \sum j = 1^D \lambda_j - \lambda_1 = \sum_{j=2}^{D} \lambda_j. \qquad (24)$$

Assuming the effective rank is $m$, the terms for $j > m$ are negligible, yielding the bound $\sum_{j=2}^{m} \lambda_j$. This represents "mode collapse" where only the dominant variation is captured.

In the multi-token case with diversity ($k > 1$), let the representation consist of $k$ tokens spanning subspace $\mathcal{S}_k$. The standard reconstruction loss $\mathcal{L}_{\text{rec}}$ encourages maximizing captured variance. However, without constraints, multiple tokens may collapse into the same dominant eigenspace (e.g., $w_1 \approx w_2 \approx v_1$), failing to reduce the error significantly below the $k = 1$ case.

We invoke Lemma 3.2: minimizing $\mathcal{L}_{\text{div}}$ enforces pairwise orthogonality of the latent factors. This constrains the basis vectors $\{w_1, \ldots, w_k\}$ to be orthogonal. According to the **Eckart-Young-Mirsky Theorem**, the optimal rank-$k$ approximation of $\Sigma_X$ under orthogonality constraints is unique and corresponds to the subspace spanned by the top-$k$ eigenvectors $\{v_1, \ldots, v_k\}$.

The captured variance becomes $\sum_{j=1}^{k} \lambda_j$. Consequently, the approximation error is:

$$\mathcal{E}_{k,\text{orth}} = \sum_{j=1}^{D} \lambda_j - \sum_{j=1}^{k} \lambda_j = \sum_{j=k+1}^{D} \lambda_j. \qquad (25)$$

Comparing the error terms, since eigenvalues are non-negative and sorted:

$$\sum_{j=k+1}^{D} \lambda_j < \sum_{j=2}^{D} \lambda_j \quad \text{(for } k \geq 2 \text{ and } \lambda_2 > 0). \qquad (26)$$

Thus, enforcing diversity via $\mathcal{L}_{\text{div}}$ ensures the representation covers distinct eigen-directions (modes), strictly reducing the approximation error by the sum of the spectral energy of the covered modes $\lambda_2, \ldots, \lambda_k$.

$\square$

### A.4. Proof of Proposition 3.5

*Proof.* Let $\mathcal{Z} = \mathbb{R}^k$ be the domain of the latent representation consisting of $k$ tokens. Let the true labeling function $c^* : \mathcal{Z} \to \mathcal{Y}$ be defined by a sparse set of ground-truth factors $z^* \in \mathbb{R}^k$. Specifically, $c^*(z^*) = \text{sign}(\langle w^*, z^* \rangle)$, where the weight vector $w^*$ is $s$-sparse ($\|w^*\|_0 \leq s$), with $s \ll k$.

We analyze the sample complexity $m(\epsilon, \delta)$, defined as the minimal number of examples required to guarantee that with probability $1 - \delta$, the empirical risk minimizer has a generalization error of at most $\epsilon$. By the Fundamental Theorem of Statistical Learning, for a hypothesis class $\mathcal{H}$ with finite VC-dimension, the sample complexity satisfies:

$$m_{\mathcal{H}}(\epsilon, \delta) \in \Theta\left(\frac{\text{VC}(\mathcal{H}) + \log(1/\delta)}{\epsilon}\right). \quad (27)$$

Assume the learned representation $F'$ is perfectly disentangled (axis-aligned with $z^*$). In this regime, the target concept remains $s$-sparse. The learner can restrict the search to the hypothesis class of $s$-sparse linear separators:

$$\mathcal{H}_{\text{sparse}} = \{x \mapsto \text{sign}(\langle w, x \rangle) \mid w \in \mathbb{R}^k, \|w\|_0 \leq s\}. \quad (28)$$

The VC-dimension of this class is bounded by $O(s \log k)$ (specifically $s \log(ek/s)$). Substituting this into the sample complexity bound:

$$m_{\text{dis}} \in O\left(\frac{s \log k}{\epsilon}\right). \quad (29)$$

Assume the representation is entangled, modeled as $F' = Uz^*$, where $U \in \mathbb{R}^{k \times k}$ is a dense rotation matrix. The optimal separator in this rotated space becomes $w' = U^{-T}w^*$. Since $U$ is dense, $w'$ becomes dense ($\|w'\|_0 \approx k$) despite $w^*$ being sparse. Consequently, a sparse classifier from $\mathcal{H}_{\text{sparse}}$ would suffer from high approximation error (bias $> \epsilon$). To achieve low error, the learner must employ the class of *dense* linear separators:

$$\mathcal{H}_{\text{dense}} = \{x \mapsto \text{sign}(\langle w, x \rangle) \mid w \in \mathbb{R}^k\}. \quad (30)$$

The VC-dimension of affine hyperplanes in $\mathbb{R}^k$ is exactly $k + 1$. Thus, the lower bound on sample complexity is:

$$m_{\text{ent}} \in \Omega\left(\frac{k}{\epsilon}\right). \quad (31)$$

Comparing the two regimes in the limit of high token count $k$ and fixed sparsity $s$:

$$\frac{m_{\text{dis}}}{m_{\text{ent}}} \propto \frac{s \log k}{k}. \quad (32)$$

Since $s \ll k$, we have $m_{\text{dis}} \ll m_{\text{ent}}$. This confirms that enforcing disentanglement transforms the learning problem from one that scales linearly with the bottleneck size ($k$) to one that scales logarithmically, strictly improving learnability in the low-data regime. $\square$

### A.5. Proof of Theorem 3.6

**Definition A.1** (Latent Gram Matrix). For a latent representation matrix $F = [f'_1, \ldots, f'_k] \in \mathbb{R}^{d \times k}$ consisting of

$k$ tokens, let the Gram matrix $G \in \mathbb{R}^{k \times k}$ be defined by $G_{ij} = \langle f'_i, f'_j \rangle$. We assume the tokens are standardized such that $\|f'_i\|_2 = 1$ for all $i \in [k]$.

**Definition A.2** (Noise Sensitivity). Let $w \in \mathbb{R}^d$ be a linear readout vector required to produce a specific target activation pattern $y \in \mathbb{R}^k$ on the latent tokens (satisfying $F^\top w = y$). Let $\eta \in \mathbb{R}^d$ be a noise vector drawn from an isotropic distribution with mean 0 and variance $\sigma^2 I$. The *Noise Sensitivity* of the readout is defined as the variance of the prediction under noise:

$$\mathcal{S}(w, F) \triangleq \text{Var}_\eta[w^\top(f'_i + \eta)]. \quad (33)$$

*Proof.* The proof proceeds in two steps: first, establishing the relationship between sensitivity and the norm of the weights; second, determining the weight norm required to resolve the worst-case target pattern $y$ under the geometry induced by $G$.

The predictor output on a perturbed input is:

$$w^\top(f'_i + \eta) = w^\top f'_i + w^\top \eta. \quad (34)$$

Since $\eta$ is zero-mean isotropic noise, the variance is determined solely by the noise term:

$$\text{Var}(w^\top \eta) = \mathbb{E}\left[\left(\sum_{j=1}^d w_j \eta_j\right)^2\right]. \quad (35)$$

Given the independence of noise entries, cross-terms vanish, yielding:

$$\mathcal{S} = \sum_j w_j^2 \text{Var}(\eta_j) = \sigma^2 \|w\|_2^2. \quad (36)$$

Thus, minimizing sensitivity is equivalent to minimizing the norm of the readout vector $w$.

We now calculate the minimum norm $w$ required to satisfy the readout constraint $F^\top w = y$. This is an underdetermined linear system (assuming $d > k$). The minimum norm solution is given by the pseudo-inverse:

$$w = F(F^\top F)^{-1}y = FG^{-1}y. \quad (37)$$

Substituting this solution into the norm calculation:

$$\begin{aligned} \|w\|_2^2 &= (FG^{-1}y)^\top(FG^{-1}y) \\ &= y^\top G^{-1} F^\top F G^{-1} y = y^\top G^{-1} y. \end{aligned} \quad (38)$$

To analyze the worst-case robustness, we consider the target pattern $y$ (with unit norm $\|y\|_2 = 1$) that maximizes this quantity. Let the eigendecomposition of $G$ be $V \Lambda V^\top$. The quadratic form is maximized when $y$ aligns with the

eigenvector corresponding to the largest eigenvalue of $G^{-1}$, which is the reciprocal of the smallest eigenvalue of $G$:

$$\max_{\|y\|=1} \|w\|_2^2 = \lambda_{\max}(G^{-1}) = \frac{1}{\lambda_{\min}(G)}. \qquad (39)$$

Substituting this into the sensitivity result from Eq 36:

$$\mathcal{S}_{\text{worst-case}} = \frac{\sigma^2}{\lambda_{\min}(G)}. \qquad (40)$$

If the tokens are orthogonal, $G = I$ and $\lambda_{\min}(G) = 1$. If tokens are entangled (correlated), $G$ is positive semi-definite with unit diagonal elements ($G_{ii} = 1$), implying $\sum \lambda_i = \text{Tr}(G) = k$. By property of the arithmetic trace, if $G \neq I$, the eigenvalues must spread around the mean of 1; therefore, there exists at least one $\lambda_j > 1$ and consequently $\lambda_{\min} < 1$. Thus, $\frac{1}{\lambda_{\min}} > 1$, strictly increasing the necessary weight norm and the resulting noise sensitivity compared to the orthogonal case. □

## B. Dataset Descriptions

*Table 8.* Summary of the processed datasets used in our experiments. The table reports the number of subjects, samples, classes, channels, timestamps per sample, patch size we use for experiments, and modality.

| Dataset | #Subject | #Sample | #Class | #Channel | #Timestamps | Patch Size | Modality |
|---|---|---|---|---|---|---|---|
| ADFTD | 88 | 69,752 | 3 | 19 | 256 | 8 | EEG |
| PTB | 198 | 64,356 | 2 | 15 | 300 | 10 | ECG |
| PTB-XL | 17,596 | 191,400 | 5 | 12 | 250 | 10 | ECG |
| APAVA | 23 | 5,967 | 2 | 16 | 256 | 8 | EEG |
| FLAAP | 8 | 530,000 | 10 | 6 | 100 | 10 | IMU (Acc+Gyro) |
| UCIHAR | 30 | 10,299 | 6 | 9 | 128 | 32 | IMU (Acc+Gyro) |
| Sleep-EDF | 100 | 42,308 | 5 | 1 | 3000 | 60 | EEG |

(1) **ADFTD** (Miltiadous et al., 2023b;a) is an EEG dataset with a three-class label for each sample, categorizing the subject as Healthy, having Frontotemporal Dementia (FTD), or Alzheimer's disease (AD). (2) **PTB** (PhysioBank, 2000) is an ECG dataset where each sample is assigned a binary label indicating whether the subject has Myocardial Infarction. (3) **PTB-XL** (Wagner et al., 2020) is a large-scale ECG dataset with a five-class diagnostic label per sample, representing a variety of cardiac conditions. (4) **APAVA** (Escudero et al., 2006) is a public EEG time series dataset with 2 classes and 23 subjects, including 12 patients with Alzheimer's disease and 11 healthy control subjects. (5) **FLAAP** (Kumar & Suresh, 2022) is a smartphone-based human-activity dataset with ten activity classes, containing six-channel IMU signals sampled at 100 Hz. (6) **UCIHAR** (Anguita et al., 2012) is a smartphone-based human-activity dataset with six activity classes, containing nine-channel IMU signals sampled at 128 Hz.(7) **Sleep-EDF** (Kemp, 2013) is a public EEG time series dataset with 100 subjects and 5 classes, which is used for the automatic detection of sleep stages.

We used patient-wise splitting protocols for all datasets to prevent data leakage. Regarding the allocation of training,validation, and testing sets, we follow the protocol proposed by Medformer(Wang et al., 2024b).

## C. Concrete Example of Disease Identification

To further demonstrate the clinical interpretability of our framework, we present a case study on Right Bundle Branch Block (RBBB). RBBB is a conduction abnormality caused by delayed right ventricular depolarization that produces a distinctive ECG morphology. According to standard cardiology references (MedicTests, 2025), an "M-shaped" QRS complex in lead V1 is a hallmark indicator of RBBB.

Using an RBBB-positive recording, we examine the fingerprint activations learned by our model. As shown in Figure 5, the most dominant fingerprint assigns high importance to physiologically meaningful regions, including the Q, R, and S deflections. These highlighted segments align with the canonical RBBB morphology, indicating that the model grounds its prediction in clinically relevant temporal structures rather than spurious correlations.

This example illustrates how our physiological fingerprints provide transparent and traceable evidence for model decisions, offering an interpretable bridge between unsupervised representation learning and established domain knowledge.

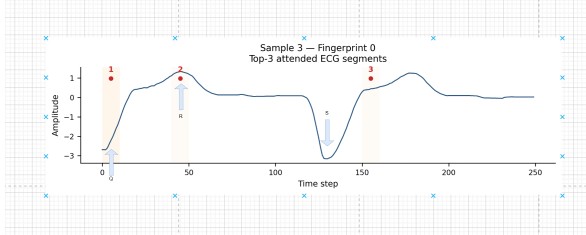

*Figure 5.* Example of fingerprint-driven interpretability on an RBBB-positive ECG sample (lead V6 shown). The highlighted patches correspond to high fingerprint activation on QRS components.

## D. Synthetic Motif-Based Dataset

To systematically analyze the interpretability and functional specialization of fingerprint tokens, we construct a controlled synthetic time-series dataset with explicit, programmatically defined temporal perturbations. Unlike real-world physiological signals, this dataset provides known ground-truth structure, allowing us to probe how different fingerprint tokens respond to salient temporal distortions.

## D.1. Base Signal Generation

Each sample is a univariate time series of length $T = 1000$. The base signal is shared across all classes and is designed to resemble smooth, low-frequency physiological dynamics. Specifically, the base signal is generated as the superposition of: (i) a low-frequency sinusoidal trend, (ii) a stochastic drift term obtained via a normalized random walk, and (iii) additive Gaussian noise. This base component ensures that classification cannot rely on trivial global statistics and instead requires localizing perturbation-dominant regions.

## D.2. Perturbation Motifs

On top of the base signal, we inject a small number of localized temporal perturbations (motifs). Each perturbation corresponds to a distinct type of signal distortion:

- **Drop:** a sustained negative-amplitude segment simulating signal dropout or abrupt suppression;

- **Oscillation:** a mid-frequency oscillatory segment of limited temporal extent.

For each sample, exactly one perturbation type is designated as the *dominant motif* according to its class label. The dominant motif is injected multiple times at random temporal locations, while other motif types are absent, ensuring clear attribution between perturbation structure and class identity.

## D.3. Class Semantics and Learning Objective

Importantly, class labels in this dataset are defined by the presence and temporal localization of perturbations, rather than by subtle differences in global signal statistics. As a result, the optimal classification strategy requires identifying and attending to perturbation-dominant regions, rather than modeling the full background dynamics.

This design intentionally encourages the emergence of a perturbation-sensitive fingerprint token, while allowing remaining tokens to capture smoother background structure. Consequently, a single fingerprint may consistently dominate class prediction, which reflects functional specialization rather than representational collapse.

## D.4. Usage in Experiments

The synthetic dataset is used in two stages. First, the TSFingerprint encoder is pre-trained in a self-supervised manner using only the reconstruction and orthogonality objectives. Second, the fingerprint representations are frozen, and a lightweight attention-based classifier is trained on top of the fingerprint tokens. This setup allows us to isolate the role of fingerprint tokens in downstream decision-making and to analyze how token-level attention aligns with injected perturbation regions.

## E. Limitations and Future Work

While TS-Fingerprint establishes a rigorous framework for disentangled representation learning, our current implementation relies on a standard linear patch-based embedding layer to tokenize the raw input. This design, while effective for capturing global dependencies, does not explicitly model multi-granularity temporal shifts or frequency-specific features at the input level, as seen in recent specialized backbones like Medformer (Wang et al., 2024b). For complex pathologies requiring micro-structure analysis (e.g., subtle high-frequency oscillations in EEG), this "naive" embedding may limit the encoder's sensitivity. Future work will explore integrating our information-theoretic bottleneck with more sophisticated hierarchical embedding modules to further enhance the resolution of the learned digital biomarkers.

## F. Disclosure of Generative AI Usage

Figure 1 (Top) was initially hand-drawn and then refined with AI to correct the details. Besides that, LLM tools are strictly limited to only automate grammar checks and word autocorrect.

