# OpenReview forum: "Learning Fingerprints for Medical Time Series with Redundancy-Constrained Information Maximization"
_ICML.cc/2026/Conference — ICML 2026 regular_

### Official Review · Reviewer_xEWu · 2026-02-13

**Soundness:** 2
**Presentation:** 3
**Significance:** 3
**Originality:** 2
**Overall Recommendation:** 4
**Confidence:** 3

**Summary:**

Paper presented "TS-Fingerprint," a self-supervised framework that compresses variable-length medical time series into a fixed-size set of k disentangled "Fingerprint Tokens" via a cross-attention bottleneck. The paper formulates the representation problem as a Redundancy-Constrained Information Maximization task, utilizing a Total Coding Rate (TCR) objective to geometrically force tokens toward statistical orthogonality. The authors claim that this approach shifts the paradigm from "entangled" masked autoencoders to a structured latent space where independent tokens specialize in distinct physiological factors. Extensive experiments across ECG and EEG benchmarks indicate that the method marginally achieves state-of-the-art classification performance while maintaining high inference efficiency. Furthermore, the paper provides mechanistic visualizations and theoretical proofs to demonstrate that the learned tokens enhance clinical interpretability and provide superior sample efficiency in low-data regimes.

**Compliance With Llm Reviewing Policy:**

Affirmed.

**Key Questions For Authors:**

To truly validate your claim that Fingerprint Tokens are "clinically self-sufficient" and disentangled, can you provide results for a standard frozen linear probe (without the task-specific attention pooling)? This would clarify whether the performance stems from the representation geometry or the additional learned pooling mechanism.

Given your goal of proposing a "foundation representation," why were recent Time Series Foundation Models (e.g., Chronos, or TimesFM) or modality-specific pre-trained models (e.g., CLOCS for ECG) excluded from the comparison?

In Figure 3, the attention allocation (Top) for FP1 appears nearly identical for both "Clean" and "Oscillation" classes. Furthermore, the local attention overlays (Bottom) show significant inconsistency: FP0 fails to attend to the first "Drop" in the signal, and FP1 attends only to the second "Oscillation." How does this inconsistent behavior prove functional specialization rather than the model simply picking up on random local shortcuts?

Regarding Figure 4, why should these attention maps be viewed as convincing evidence of captured physiological factors? The class-wise distributions appear static across different labels, and for ECG, multiple tokens redundantly highlight the same QRS peaks. Please clarify how this demonstrates disentanglement or any specific mapping to clinical biomarkers beyond generic saliency.

**Limitations:**

No. While the authors include a "Limitations" section, it is technically narrow and fails to address the broader implications of their "Fingerprint" paradigm.

**Strengths And Weaknesses:**

Strengths

1.	Well-motivated problem framing. The paper clearly argues that MAE-style pretraining can yield representations that are effective but not explicitly compact or semantically interpretable, often relying on heuristic aggregation (pooling/ CLS ). It then positions “fingerprint tokens” as a direct remedy: a fixed-size representation designed to be both sufficient and more interpretable for clinical signals.
2.	Clean objective design that targets sufficiency and redundancy reduction explicitly. TS-Fingerprint uses a simple dual objective: reconstruction to encourage sufficiency, plus a TCR-based diversity penalty to minimize redundancy and push tokens toward statistical disentanglement. The method section also motivates the diversity term as a concrete way to avoid feature collapse and enforce “repulsion” among tokens.
3.	Meaningful theoretical grounding that makes the geometry of the diversity term explicit. The paper provides a clear geometric interpretation: optimizing the TCR/diversity loss corresponds to maximizing latent volume and (under stated assumptions) encourages orthogonality/diagonal covariance among tokens. It also connects token geometry to robustness via a noise-sensitivity analysis based on the latent Gram matrix, highlighting why orthogonality can improve stability.
4.	Broad experimental coverage with substantial baseline breadth. The evaluation spans multiple ECG/EEG datasets (and other modalities), with an explicit patient-wise split protocol to prevent leakage. They also benchmark against a large suite (noted as 11 baselines) and report aggregate ranking across datasets/metrics, which helps contextualize performance beyond one dataset.
5.	Competitive performance with an explicit efficiency claim. The paper reports strong overall average-rank performance across tasks and emphasizes that the approach retains fast inference. Concretely, it claims ~14% faster inference than Medformer while maintaining top aggregate rank, which is practically relevant in clinical pipelines.
6.	Ablations (Tables 2 & 3) support that the redundancy-constrained objective adds value beyond reconstruction alone. The pretraining ablation compares training from scratch vs. pretraining with (L_{rec}) vs. (L_{rec}+L_{div}), showing consistent improvements and attributing part of the gain to the diversity term. The paper also reports sensitivity analysis over token count and masking ratio, suggesting the method has a reasonably stable operating region rather than requiring brittle tuning.


Weaknesses

1.	The method combines a Perceiver-style cross-attention bottleneck that compresses variable-length inputs into (k) learnable tokens with a Total Coding Rate (TCR)–based diversity penalty and a masked reconstruction decoder. While this is a sensible and well-motivated integration, the paper repeatedly frames it as a “paradigm shift,” but it reads more like a careful adaptation of existing redundancy-reduction theory (TCR) to a standard latent-token bottleneck. A clearer delineation of what is fundamentally new beyond reusing TCR in this architecture (and beyond prior latent bottlenecks), and toning down of the claims would much strengthen the contribution.
2.	The introduction criticizes prior MAE-style representations for relying on aggregation heuristics such as global pooling or a designated ([CLS]) token. However, downstream inference introduces a task-specific query and attention pooling over fingerprint tokens ((q_{\text{task}}), (\alpha_i)) to form the final representation. If the main claim is that the learned tokens are independently useful/disentangled, the paper should report standard frozen linear-probe (or KNN) results directly on the concatenated tokens to avoid conflating representation quality with an additional learned pooling mechanism. This is standard practice in paper which introduce foundation models. The representations need to be useful on their own.
3.	Empirical improvements over strong baselines are modest (though significant statistically), but the framing is overstated. On key benchmarks, gains over specialized models can be relatively small (e.g., PTB-XL F1 improves from Medformer’s ~62.0 to ~63.5). At the same time, the paper uses emphatic language (“superiority,” “paradigm shift,” “massive” promotion) that may not match the absolute margins in several settings. Again, dialing back rhetoric and emphasizing where the method helps most (and where it does not) would make the claims feel more proportionate to the results, and contribute to this paper.
4.	The “ideal clinical representation” axioms are overly restrictive and not scoped to task types. The paper asserts an ideal representation must be compressed into a fixed-size vector invariant to input length, along with sufficiency and disentanglement. This assumption fits patient-level classification, but it is much less appropriate for dense prediction settings common in medical time series (segmentation, event detection, forecasting), where time-local outputs matter. The paper would benefit from explicitly scoping the “ideal” criteria to the classification/phenotyping regime and discussing limitations for dense tasks.
5.	The theoretical story relies on strong distributional assumptions and may be heavier than what is empirically validated. The equivalence between the composite objective and minimizing Total Correlation is derived under an explicit Gaussian assumption on the latent distribution. Yet the paper does not empirically test whether learned token distributions for complex MedTS are even approximately Gaussian, which weakens the practical force of the argument. Additionally, several theoretical results read as expected consequences of enforcing orthogonality/low-redundancy (via TCR) rather than unique, method-specific insights.
6.	Mechanistic “disentanglement” evidence is largely qualitative and vulnerable to cherry-picking. Figure 3 is presented as “direct mechanistic evidence” that different tokens specialize in different perturbations, but it is based on a synthetic perturbation dataset and primarily visual inspection. This makes the figure closer to a sanity check than a convincing disentanglement demonstration. Stronger support would include dataset-wide quantitative localization/alignment metrics plus comparisons to ablated baselines (no (L_{div}), standard bottleneck, etc.).
7.	“Adaptive sparsity” may reflect dataset-level fixed token weighting rather than per-input dynamic allocation. The paper interprets class-wise attention distributions as an intrinsic mechanism that adapts token usage to modality complexity (ECG vs EEG). However, the presented heatmaps appear broadly similar across classes within a dataset, which is consistent with a largely static, dataset-specific weighting learned by the classifier rather than input-conditional routing. The work would be stronger with analyses showing per-sample variability (and stability across seeds), and with evidence that token usage changes meaningfully with pathology type rather than only by dataset.
8.	Token “division of labor” is not fully reconciled with apparent redundancy in attended regions. The interpretability narrative emphasizes that different tokens capture independent physiological factors and become orthogonal/disentangled. Yet the attention-window visualizations can suggest multiple tokens focusing on highly salient, overlapping waveform events (e.g., prominent ECG complexes), which may indicate redundancy or shared shortcuts rather than clean specialization. A more systematic redundancy analysis (overlap statistics across tokens, mutual information, or token-drop ablations) would help reconcile these observations with the claimed disentanglement.
9.	“Foundation representation” framing is not convincingly supported by the comparison set. The experiments claim the goal of a “foundation representation” and compare against several transformers and a medical specialist model, plus two SSL baselines. However, the evaluation does not situate TS-Fingerprint against newer large-scale time-series foundation models (like Chronos, TimesFM, etc..) or modality-specific pre-trained models (there are a lot of ECG models for example, and other SSL papers on ECG e.g CLOCS, by D Kiyasseh · 2020 ), which is important if the paper wants to argue broadly general “foundation” status. Without those comparisons (or at least a clear justification for their omission), the scope of the claim feels ahead of what the experiments establish.
10.	Presentation issues: conceptual art and repetitive phrasing reduce clarity and perceived rigor. Figure 1 positions the method as a “paradigm shift” using stylized conceptual graphics, but the visual metaphor is not especially helpful for understanding the concrete architecture/objective and can feel distracting. The paper also discloses AI refinement for Figure 1, and the figure’s style does not match the rest of the technical presentation. Replacing the conceptual art with a clean schematic of the bottleneck + objectives (and tightening repeated rhetorical phrases like “geometric regularization” / “paradigm shift”) would improve readability and credibility.

---

> ### Author Rebuttal · Authors · 2026-03-26
>
> ## w1, w3, w4. Toning down of the claims
> We agree that several claims in the current draft are broader than what the evidence supports. In revision, we will narrow the framing from “paradigm shift” / “foundation representation” to a more specific contribution: a structured fixed-size representation for patient-level medical time-series classification. We will also scope the “ideal clinical representation” discussion to classification/phenotyping rather than dense prediction. As for the performance margin, given that our model is more efficient and lightweight than other models, the gap remains substantial.
>
> ## w2, Q1. linear-probe
> On ADFTD, a linear probe on frozen pretrained Fingerprint Tokens improves over scratch (Accuracy: 48.83→51.70, F1: 45.80→46.51), showing that the representation itself is useful even without task-specific attention pooling. We will add this result and clarify the distinction between representation quality and readout design.
>
> ## w5. Strong distributional assumptions
> We agree that the theoretical interpretation should be stated more carefully.
> Let $Z = F'$ with covariance $\Sigma$. Define the Gaussian surrogate
> $$
> \widetilde{TC}(Z) := \frac{1}{2}\log \frac{\prod_{i=1}^k \Sigma_{ii}}{\det(\Sigma)}.
> $$
> If $Z$ is Gaussian, then this quantity is exactly the total correlation $TC(Z)$. If $Z$ is not Gaussian, $\widetilde{TC}(Z)$ is still well-defined from $\Sigma$ alone.
>
> By Hadamard's inequality,
> $$
> \det(\Sigma) \leq \prod_{i=1}^k \Sigma_{ii},
> $$
> so $\widetilde{TC}(Z) \geq 0$, with equality if and only if $\Sigma$ is diagonal. Therefore, under fixed token energy, maximizing the log-determinant drives the covariance toward a diagonal form, i.e., decorrelation and orthogonality in expectation. This is a matrix-geometric fact about covariance structure and does not require the latent law to be exactly Gaussian once the covariance matrix is defined.
>
> The same observation extends to the later guarantees. Proposition 3.3 is a spectral approximation result: orthogonality ensures coverage of the top-$k$ eigendirections and reduces reconstruction error to the residual tail energy. Theorem 3.6 is a robustness result stated entirely in terms of the Gram matrix $G = F^\top F$: the worst-case noise sensitivity scales as
> $$
> \frac{\sigma^2}{\lambda_{\min}(G)},
> $$
> and this quantity is minimized when the tokens are pairwise orthogonal, i.e., when $G = I$. Neither statement requires an exact Gaussian latent distribution; both depend only on the second-order geometry of the learned representation.
> Thus, Gaussianity is only needed for the exact total-correlation identity. Beyond Gaussianity, $L_{\mathrm{div}}$ remains a principled covariance-volume surrogate that still enforces the geometric properties used in our approximation and robustness results. We will revise the wording accordingly and avoid presenting the Gaussian case as a universal requirement.
>
> ## w6, w7, w8, Q3, Q4. Token redundancy vs. division of labor
> We agree that the visualizations alone are not sufficient to establish disentanglement. We added quantitative redundancy analysis on ADFTD with
> $K=8$.
> | | Attn CosSim ↓ | Attn Top-20% IoU ↓ | FP CosSim ↓ | SVD Eff. Rank ↑ (max=8) |
> |---|---|---|---|---|
> | w/ $L_\text{div}$ | **0.08** | **0.10** | **0.02** | **7.30** |
> | w/o $L_\text{div}$ | 0.22 | 0.28 | 0.23 | 6.17 |
>
> Removing $L_{div}$ increases redundancy and lowers effective rank. These results support specialization rather than collapse, and we will add them in the revision.
>
> ## w9, Q2. Foundation model baseline and other SSL papers
> We additionally compared TS-Fingerprint with a recent foundation-style time-series model (GPT4TS and TimesFM) and modality-specific SSL baselines on ADFTD:
>
> | | GPT4TS [1] | TimesFM [2]  | CLOCS [3] | MTS-LOF [4] | Ours |
> |---|---|---|---|---|---|
> | Accuracy | **54.11 ± 0.67** | 52.77±1.71 | 49.31 ± 0.88 | 52.07 ± 0.77  | 53.91 ± 0.54 |
> | Precision | 47.76 ± 0.94 | 46.89±1.2 | 46.86 ± 0.74 | 45.63 ± 0.71 | **51.89 ± 0.49** |
> | Recall | 46.63 ± 0.87 | 46.34±0.53 | 43.67 ± 0.81  | 50.52 ± 0.93 | **52.10 ± 1.09** |
> | F1 | 44.66 ± 0.58 | 45.25±1.05  | 45.21 ± 0.901 | 47.95± 0.82 | **51.79 ± 0.74** |
> | AUROC | 67.86 ± 0.52 | 66.17±0.99 | 67.31±0.84 | 68.89±0.772 | **72.11 ± 0.73** |
>
> ## w10. Presentation issues
> We agree that the current presentation is more rhetorical than necessary. In revision, we will simplify or replace Figure 1 with a cleaner technical schematic, reduce repeated high-level phrasing, and use more proportionate wording throughout the introduction, experiments, and conclusion.
>
> [1] Tian et al. "One fits all: Power general time series analysis by pretrained lm." Neurips 2023.
>
> [2] Das et al. "A decoder-only foundation model for time-series forecasting." 2024.
>
> [3] Kiyasseh et al. "Clocs: Contrastive learning of cardiac signals across space, time, and patients." ICML 2021.
>
> [4] Li et al. "MTS-LOF: medical time-series representation learning via occlusion-invariant features." JBHI 2024.

---

> > ### Author Rebuttal · Reviewer_xEWu · 2026-04-02
> >
> > Thank you for your rebuttal. I maintain my rating for the paper.

---

### Official Review · Reviewer_cA98 · 2026-02-16

**Soundness:** 3
**Presentation:** 3
**Significance:** 3
**Originality:** 3
**Overall Recommendation:** 5
**Confidence:** 4

**Summary:**

The paper proposes a novel self-supervised learning framework designed specifically for Medical Time Series. The authors argue that existing MAE-based approaches for time series produce "entangled" representations that conflate noise, identity, and pathology, resulting in suboptimal diagnostic performance. To address this, they introduce a "Fingerprint" architecture that compresses variable-length signals into a fixed set of latent tokens via a cross-attention bottleneck. The training objective combines a masked reconstruction loss with a diversity loss. Overall, the authors present the concept of treating the encoder as a "Redundancy-Constrained Information Maximizer" to generate compact, interpretable, and disentangled representations.

**Compliance With Llm Reviewing Policy:**

Affirmed.

**Final Justification:**

The authors addressed my concerns, I maintain my score.

**Key Questions For Authors:**

1. Have the authors experimented with different downstream aggregation strategies for the fingerprint tokens? Currently, the paper uses an attention pooling mechanism (Eq. 4). Does a simple flattening or mean pooling work, or is the attention mechanism crucial for selecting the "active" fingerprint for a specific diagnosis?
2. How does the method handle artifacts/noise that are distinct but irrelevant (e.g., electrode pop in EEG)? Does the model assign a specific "noise token" to these artifacts, or are they filtered out by the bottleneck?

**Limitations:**

1. The current architecture assumes regularly sampled time series that can be cleanly patched. It does not address the common clinical reality of sparse or irregularly sampled data (e.g., EHR time series), limiting its immediate applicability to broader electronic health record data without pre-processing/interpolation.

**Strengths And Weaknesses:**

### Strengths
1. The manuscript proceeds to analyze the concept of representation learning through a rigorous Disentangled Rate-Distortion framework. The derivation linking the minimization of TCR to the minimization of Total Correlation (TC) and the enforcement of orthogonality provides a strong mathematical justification for the proposed diversity loss.
2. The shift from standard sequence-to-sequence pre-training to a "Fixed-Rank Bottleneck" using learnable queries is a clever inductive bias. By forcing the model to compress information into a small, fixed number of slots and penalizing redundancy, the model is structurally encouraged to isolate independent factors of variation.
3. The method demonstrates superior performance across a wide range of physiological modalities (EEG, ECG, IMU). Notably, it outperforms both general time-series SOTA (PatchTST, iTransformer) and domain-specific models (Medformer). The significant gains on the complex ADFTD dataset (Alzheimer's detection via EEG) highlight the method's ability to capture subtle pathological features that holistic models miss.

### Weaknesses
1. As acknowledged in the Limitations section, the model uses a standard linear patch embedding. Given the multi-scale nature of physiological signals (e.g., high-frequency ripples vs. slow waves in EEG), the lack of a multi-granularity embedding (as seen in Medformer or MTST) might be a bottleneck. It would be interesting to see if integrating multi-scale patching would further enhance the "fingerprint" quality.
2. While the ablation study (Table 3) shows robustness for the hyperparameter $k$, there is no theoretical guide for selecting $k$ for a new, unknown dataset. If the underlying physiological manifold has a dimensionality much higher than $k$, the bottleneck might discard relevant pathological information (Proposition 3.3).

---

> ### Author Rebuttal · Authors · 2026-03-26
>
> ## w1. Multi-granularity embedding
> We agree that integrating multi-scale input embeddings is a natural extension. Our current implementation uses a standard linear patch embedding, which is effective but does not explicitly model multi-granularity temporal shifts or frequency-specific structure. This is consistent with the limitation already discussed in the paper, where we note that stronger hierarchical or multi-scale embedding modules could further improve the quality of the learned fingerprints. We tested our framework with a Medformer-style multi-scale input embedding on ADFTD. The preliminary results are:
>
> | | vanilla embeddings  | multi-scale embeddings |
> |---|---|---|
> | Accuracy | 53.91 | **55.06** |
> | F1 |  51.79 | **53.01** |
>
> We will include the full results and discussion in the revision.
>
> ## w2. Hyperparameter k
> Unfortunately, we can not claim a universal closed-form rule for a new dataset. Our current view is that Proposition 3.3 provides qualitative guidance: increasing k allows the bottleneck to cover more independent directions, reducing the approximation-error bound from $\sum_{j=2}^{D}\lambda_j$ to $\sum_{j=k+1}^{D}\lambda_j$, while an overly large $k$ can introduce redundancy and weaken the division of labor across tokens. Empirically, Table 3 shows that the method is reasonably robust across different choices of k with the best trade-off at k=8 on PTB-XL.
>
> ## q1. Downstream aggregation strategies
> We chose attention pooling because it is aligned with our formulation of clinical inference. In Eq. (4), the task query selectively reads out the fingerprint tokens, and Proposition 3.5 shows why a selective sparse readout is well matched to pathology detection driven by a small subset of factors. We also compared attention pooling against simpler alternatives:
> | | Mean pooling | Flattenning | Attention Pooling |
> |---|---|---|---|
> | Accuracy | 51.67 | 53.63 | 53.91 |
> | F1 | 48.83 | 49.83 | 51.79 |
>
> We will clarify this motivation and include full comparisons with other aggregation strategies.
>
> ## q2. Noise token
> Our method does not explicitly enforce a dedicated “noise token.” The intended mechanism is indirect: the fixed-rank bottleneck limits capacity, the reconstruction term preserves sufficient information, and the diversity term encourages different tokens to specialize rather than collapse. In this sense, some unstable or diagnostically irrelevant noise may be suppressed by the bottleneck, while some persistent but irrelevant artifacts may still be encoded in the fingerprint tokens and then down-weighted by the downstream attention pooling. This interpretation is also consistent with our perturbation analysis and with Theorem 3.6, which links more orthogonal token geometry to stronger robustness against uncorrelated noise.
>
> ## lim1. Irregularly sampled signals
> We agree that the current framework is mainly designed for regularly sampled signals that can be patched directly. Extending the framework to sparse or irregularly sampled clinical time series such as EHR data would require additional design at the tokenization such as multi-scale embeddings, and we view this as an important direction for future work.

---

> > ### Author Rebuttal · Reviewer_cA98 · 2026-04-02
> >
> > Thanks for the clarifications, I will keep my score.

---

### Official Review · Reviewer_7aYs · 2026-03-12

**Soundness:** 3
**Presentation:** 3
**Significance:** 3
**Originality:** 3
**Overall Recommendation:** 5
**Confidence:** 5

**Summary:**

This paper studies the intrinsic properties of medical time series (MedTS) and proposes a tailored representation learning framework. The method aims to disentangle entangled factors in ECG signals, including signal identity, pathology, and noise, by learning k orthogonal fingerprint tokens using a redundancy-constrained information maximization objective. The model is pre-trained with masked reconstruction and token orthogonality losses. Extensive theoretical analysis and experiments are provided, and the proposed method achieves state-of-the-art performance with the best average ranking among existing methods.

**Compliance With Llm Reviewing Policy:**

Affirmed.

**Key Questions For Authors:**

See weakness

**Limitations:**

See weakness

**Strengths And Weaknesses:**

**Strengths:**
The motivation of the proposed method is well grounded in the inherent characteristics of medical time series (MedTS). The theoretical formulation is comprehensive and well-developed, and the overall methodology is logically presented. The figure illustrating the method pipeline is clear and informative, providing sufficient details for understanding the framework. The experimental section is thorough and includes adequate implementation details. For example, the interpretability analysis presented in Figure 4 is interesting and insightful. From my experience, EEG signals are generally more difficult to disentangle than ECG signals, especially for datasets such as ADFTD that are used for brain disease detection. Overall, this paper presents a complete study with sufficient novelty.

---

**Weaknesses:**
There are no major weaknesses in this paper, but I do have several suggestions and questions for the authors:

1) In line 208, the method maintains orthogonality by minimizing the negative log-determinant (referred to as the diversity loss). Why was this specific formulation chosen? Are there alternative formulations that could also enforce orthogonality? Have the authors conducted any preliminary experiments comparing different orthogonality constraints?

2) The method adopts a pre-training and fine-tuning pipeline, where the diversity loss is applied during pre-training. Have the authors tried training the model from scratch with the diversity loss applied directly? As shown in Table 4, the model trained from scratch on ADFTD achieves an F1 score of 45.80%, while pre-training improves it to 51.79%. It is therefore somewhat unclear whether the performance gain mainly comes from the diversity loss or from the pre-training itself. The comparisons with Ti-MAE and SimMTM are not entirely convincing since they use different backbone architectures. If the diversity loss can be applied directly during training from scratch, this method could serve as a plug-and-play module that can be integrated with many other methods, making it even more appealing in practice.

3) What backbone architecture is used in this method? It would be helpful to add a short paragraph describing the backbone in more detail, including how the input is embedded and whether any specific modules are used within the Transformer architecture.

4) In Figure 4, does the yellow color indicate a value close to zero or a very low score? The attention pooling mechanism in line 227 should not produce a 0 value.

---

> ### Author Rebuttal · Authors · 2026-03-26
>
> ## w1. Diversity loss
> We chose the negative log-determinant objective because it is aligned with our geometric goal: it maximizes the volume spanned by the fingerprint tokens in latent space. Under the fixed-energy assumption in Lemma 3.2, this corresponds to reducing redundancy and encouraging orthogonality across tokens.
>
> We agree that alternative decorrelation-based formulations are possible. To validate this design choice, we additionally implemented VICReg covariance loss [1] and Barlow Twins loss [2] as alternatives.
>
> We quantitatively compare the off-diagonal correlation of the learned token representations:
>
> | Method                     | Off-diagonal Mean Correlation ↓ | Accuracy ↑ | F1 ↑ |
> |--------------------------|--------------------------------|------------|------|
> | No constraint            | 0.897                          | 53.51    |   50.48
> | Barlow Twins [2]         | 0.783                          |     49.50      | 44.62    |
> | VICReg (covariance) [1]  | 0.637                          | 49.87          |46.47     |
> | **Ours (log-det)**       | **0.022**                      | **53.91**  | **51.79** |
>
> These results demonstrate that our objective enforces substantially stronger decorrelation than other methods, leading to near-orthogonal and more disentangled token representations.
>
> ## w2. Diversity regularization in pretraining/training.
> We will add a controlled comparison between CE-only and CE+Ldiv training from scratch. Here we showed a snapshot of results on  ADFTD:
> | scratch CE-only | scratch CE+Ldiv  | Finetuning |
> |---|---|---|
> | 48.83, 45.80 | 51.87, 49.36 | 53.91, 51.79 |
>
> Hence, it suggests not only that pretraining is beneficial, but also that the reconstruction and diversity terms play complementary roles, with $L_{rec}$ preserving sufficiency and $L_{div}$ preventing token collapse by reducing redundancy.
>
> ## w3. Backbone
> The current model uses a standard patch-based embedding, followed by a Transformer-style encoder with a learnable query set that cross-attends to patch tokens to produce a fixed set of fingerprint tokens. We will extend the main text to discuss more details of the backbone.
>
> ## w4. Figure 4
> Since the downstream weights are produced by a softmax attention pooling, the light/yellow color indicates a low but non-zero normalized attention score, not an exact zero. We will revise the caption and add a clearer legend/colorbar.
>
> [1] Bardes et al., "VICReg: Variance-Invariance-Covariance Regularization", ICLR 2022
>
> [2] Zbontar et al., "Barlow Twins", ICML 2021

---

> > ### Author Rebuttal · Reviewer_7aYs · 2026-04-03
> >
> > All my questions have been resolved.

---

### Official Review · Reviewer_yAxg · 2026-03-13

**Soundness:** 3
**Presentation:** 3
**Significance:** 3
**Originality:** 2
**Overall Recommendation:** 4
**Confidence:** 4

**Summary:**

The paper attempts to force auto-encoders to embed inputs in disentangled spaces that resemble signal factorization as opposed to mere compression. This is achieved using a combined training objective that includes both masked reconstruction loss and a geometric orthogonality loss. Results show that the work improves inference quality over a number of baselines.

**Compliance With Llm Reviewing Policy:**

Affirmed.

**Final Justification:**

The authors have addressed my concerns, expressed in the original review. I have updated my recommendations to a weak accept.

**Key Questions For Authors:**

1. How would the approach compare to solutions designed specifically for disentangled representation learning such as those survey in reference [1], mentioned in this review, above?

2. How would the approach compare to recent structure state space solutions such as Mamba and its variants?

**Limitations:**

Yes.

**Strengths And Weaknesses:**

At its core, the paper addresses the problem of disentangled representation learning. Disentangled representations are more interpretable and may simplify subsequent fine-tuning, leading to improved performance (e.g., on such downstream tasks as classification) due to the cleaner structure of the latent space. The representation is further forced to be of fixed dimensionality, designed to create a more severe bottleneck that prevents overfitting. While the approach is well written and well explained, the evaluation has several drawbacks:

1. There has been a lot of work on disentangled representation learning. The evaluation section does not compare the proposed solution to approaches that are designed specifically to reduce disentanglement in the latent space. A good recent survey on this topic is found here:

[1] Xin Wang, Hong Chen, Si'ao Tang, Zihao Wu, and Wenwu Zhu. "Disentangled representation learning." IEEE Transactions on Pattern Analysis and Machine Intelligence 46, no. 12 (2024): 9677-9696.

As such, the evaluation is somewhat unfair and not representative of the state of the art in the field.

2. The evaluation focuses more on the encoder backbone (see Table 1), and less on the training method (Table 4). Yet, the quality of latent representations produced by different representation learning schemes is a function of both encoder architecture and loss function (i.e., learning objective). It would have been good if Table 4 was expanded to explore a broader range of training frameworks.

3. While the presented solution is developed for time-series data, the evaluation does not seem to compare the approach with some of the recent popular encoder architecture designed for efficient time-series data representation, such as Mamba and its variants. Thus, it is not clear if the new approach beats the state of the art.

In view of the above lack of a comprehensive comparison, this reviewer believes that the paper could use a revision that features better coverage of competing state of the art.

---

> ### Author Rebuttal · Authors · 2026-03-26
>
> ## w1, q1. disentangled representation learning
> We agree that the current manuscript does not include enough direct comparisons to methods explicitly designed for disentangled representation learning. In the revision, we will expand the discussion of the literature covered in [1] and clarify the relation between our objective and prior disentanglement-oriented approaches. We added a controlled comparison by training the same backbone (encoder + FP tokens) with a representative disentanglement objective [2][3] on ADFTD. Under this matched-backbone setting, [2] improves over scratch, while [3] even hurts the performances. Our method remains stronger:
>
> | Method   | Accuracy ↑ | F1 Score ↑ |
> |----------|------------|------------|
> | Scratch  | 48.83      | 45.80      |
> | [2]      | 50.24      | 47.33      |
> | [3]      | 48.35      | 43.86      |
> | **Ours** | **53.91**  | **51.79**  |
>
> [2] and [3] address disentanglement in settings that are related but methodologically different from ours. [2] learns shared/exclusive representations from paired inputs using mutual information estimation, maximizing information for the shared and exclusive parts while reducing their mutual dependence, and it does so without relying on reconstruction or generation. [3] is a supervised few-shot image classification framework that explicitly separates a classification branch from a variation branch, aiming to decouple class-discriminative features from class-irrelevant factors such as background, style, or domain.
>
> In our revised manuscript, we will add more comprehensive comparison and carefully discuss [1].
>
> ## w2 w3 q2. Broader Baselines
> **For Table 4**: we will add more pretrain frameworks for medical time series such as [4, 5] to compare the performance gain brought by pretraining. Based on ADFD dataset,
> |  | [4] | [5] | Ours |
> |---|---|---|---|
> |scratch| 51.38, 45.63 | 50.66, 44.85 | 48.83, 45.80 |
> |finetune| 52.07, 47.95 | 50.15, 45.93 | 53.91 (+10.40%), 51.79 (+13.07%) |
>
> These results indicate that our model trained from scratch is not necessarily the strongest baseline. However, the key observation is that our method benefits significantly more from pretraining compared to prior approaches. Specifically, we observe substantial improvements in both Accuracy and F1 after pretraining, highlighting the effectiveness of our representation learning strategy.
>
> This is precisely the main point that Table 4 is intended to convey. We will further clarify this in the revised manuscript and include a more comprehensive comparison for completeness.
>
> **For Table 1**: we will add more recent encoder architecture such as [6, 7, 8]. The [8] is a recent mamba-based method which performs well on EEG. Under the same evaluation protocol on ADFTD, the preliminary results are:
> |            | [6]            | [7]            | [8]              | Ours              |
> |------------|----------------|----------------|------------------|-------------------|
> | Accuracy   | 48.37±4.30     | 51.85±2.92     | 49.24 ± 1.78     | **53.91±0.54**    |
> | Precision  | 47.97±4.69     | 48.75±1.11     | 46.57 ± 1.29     | **51.89±0.49**    |
> | Recall     | 46.21±4.01     | 48.25±1.37     | 46.62 ± 0.99     | **52.10±1.09**    |
> | F1 score   | 46.63±4.17     | 47.66±1.10     | 46.35 ± 1.03     | **51.79±0.74**    |
> | AUROC      | 64.82±4.33     | 68.19±0.92     | 68.73 ± 1.01     | **72.11±0.73**    |
>
> We will include the full reproduced results and expand the baseline coverage in the revised Table 1.
>
> [1] Wang et al. "Disentangled representation learning." TPAMI 2024.
>
> [2] Sanchez et al. "Learning disentangled representations via mutual information estimation." ECCV 2020.
>
> [3] H. Cheng, Y. Wang, H. Li, A. C. Kot, and B. Wen, “Disentangled feature representation for few-shot image classification,” arXiv 2021.
>
> [4] Li et al. "MTS-LOF: medical time-series representation learning via occlusion-invariant features." JBHI 2024.
>
> [5] Wang et al. "Contrast everything: A hierarchical contrastive framework for medical time-series." Neurips 2023.
>
> [6] Fan et al. "Towards multi-resolution spatiotemporal graph learning for medical time series classification." WWW 2025.
>
> [7] Yuan et al. "Reading Between the Channels: Knowledge-Augmented Medical Time Series Classification." ACMMM 2025.
>
> [8] Hong et al. "An Efficient Self-Supervised Framework for Long-Sequence EEG Modeling." arXiv 2025.

---

> > ### Author Rebuttal · Reviewer_yAxg · 2026-04-02
> >
> > The suggested added material by the authors alleviates my concerns. I will update my recommendation to a weak accept accordingly.

---

### Decision · Program_Chairs · 2026-04-30

**Decision:**

Accept (regular)

**Comment:**

This paper proposes a self-supervised framework for medical time series representation learning that produces a fixed set of “fingerprint tokens” via a cross-attention bottleneck, trained with a reconstruction objective and a redundancy-constrained diversity loss, aiming to learn compact, interpretable, and disentangled representations.

The paper addresses an important problem, and the proposed approach is well motivated and technically sound. The method is supported by a clear formulation, including a principled diversity objective based on Total Coding Rate, and demonstrates strong empirical performance across multiple datasets.

Overall, the paper presents a coherent and well-supported contribution with consistent empirical benefits, and meets the acceptance criteria.